# When Labelers Stay Silent:
# The Power of Ties in Cost-Effective Preference Learning

**Jiaqi Lv** [1 2]  **Zihan Zhang** [1 2]  **Chengjing Yang** [1 2]  **Shiyu Xia** [1 2]  **Ning Xu** [1 2 3 †]  **Xin Geng** [1 2 †]

## Abstract

Standard preference alignment relies on a binary *forced-choice paradigm*, assuming definitive preferences for all pairs. However, we find that indistinguishable pairs are prevalent even in standard benchmarks, where quality differences of two responses often fall below the labeler's discriminative resolution limit. Forcing a choice in such cases could inject significant noise that undermines policy optimization. In this work, we propose a *silent-aware framework* that introduces a principled way to allow annotators to stay silent (i.e., express ties) and then explicitly model these ties during optimization. Our findings reveal a compelling phenomenon: when ties are properly modeled, supervision from small models yields alignment surpassing that of forced-choice LLMs or human experts. This discovery highlights a *cost-effective path for alignment*: respecting a labeler's resolution limit is more critical than increasing its capability, while simultaneously *unlocking the latent value* in existing benchmarks by properly modeling inherent tie signals without requiring any re-labeling effort. To leverage these signals, we propose several optimization objectives to drive the policy toward high-reward regions while mitigating unreliable updates that lead to arbitrary distribution shifts. Our approaches transcend conventional alignment performance, consistently outperforming the strongest available baselines across diverse benchmarks.

## 1. Introduction

The remarkable evolution of large language models (LLMs) has necessitated rigorous alignment protocols to guarantee that outputs remain beneficial and ethically grounded (Askell et al., 2021; OpenAI, 2023; Team, 2023). A fundamental pillar of current alignment strategies involves optimizing model policies based on comparative feedback, typically through preference-based optimization. Central to this process is the Bradley-Terry (BT) model (Bradley & Terry, 1952), which provides the theoretical foundation for dominant frameworks like reinforcement learning from human feedback (RLHF) (Christiano et al., 2017; Bai et al., 2022a; Ouyang et al., 2022) and direct preference optimization (DPO) (Rafailov et al., 2023). This standard paradigm operates on a fundamental assumption that for any given pair of responses, *a definitive preference* can always be established. However, current annotation workflows, whether relying on human labor or automated feedback from frontier models (i.e., RLAIF (Lee et al., 2023)), overlook a critical constraint: *the discriminative resolution limit* of labelers. When the quality difference between two responses falls below a labeler's sensory threshold, a binary distinction becomes increasingly unreliable and prone to bias. Such forced-choice labels would inevitably introduce noise into the training data, ultimately degrading the robustness and efficiency of the alignment process.

To quantify the prevalence and impact of this issue, we begin by observing four standard preference datasets. We characterize *tie pairs* as those where a model assigns nearly identical rewards to both responses, suggesting the quality difference approaches the model's discriminative resolution limit. As illustrated in Figure 1, we observe that a significant portion of response pairs actually yield indistinguishable quality signals when evaluated using the implicit rewards of Llama-3B and Llama-8B (the specific labeling protocol is detailed in Sec 3.2). The frequency of these ties correlates inversely with the labeler's scale, reflecting the higher epistemic uncertainty inherent in smaller models. Notably, *for golden labelers* like GPT-4 or human annotators, indistinguishable pairs persist even under a strict criterion of exact score equality, yet these were originally labeled with forced binary preferences. These instances represent the inherent

---

[1]Southeast University, China [2]Key Laboratory of New Generation Artificial Intelligence Technology and Its Interdisciplinary Applications (Southeast University), Ministry of Education, China [3]National Center of Technology Innovation for EDA, China. Correspondence to: Ning Xu <xning@seu.edu.cn>, Xin Geng <xgeng@seu.edu.cn>.

*Proceedings of the 43rd International Conference on Machine Learning*, Seoul, South Korea. PMLR 306, 2026. Copyright 2026 by the author(s).

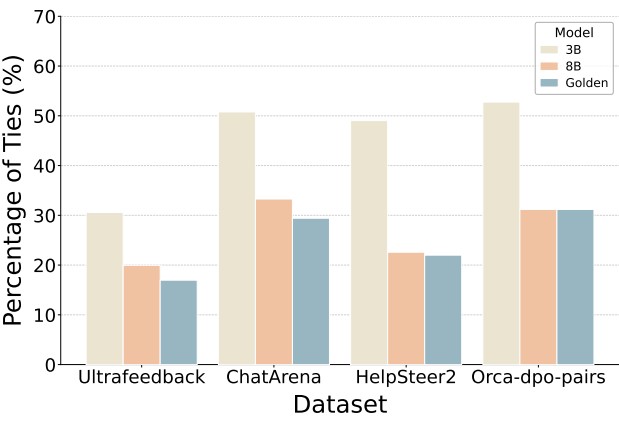

*Figure 1.* Distribution of ties across labelers of varying scales. Indistinguishable pairs are defined by a resolution threshold $\delta_{\mathcal{M}}$, applied uniformly to both Llama-3B and 8B models. "Golden" denotes the intrinsic preference and scores natively provided by the original datasets.

resolution limit of any annotator; in such cases, remaining silent about which is better may provide a more honest and reliable signal than a forced preference.

We next evaluate the impact of these tie pairs on alignment by comparing several configurations on UltraFeedback (Cui et al., 2023) (detailed settings in Section 4.1), with results summarized in Table 1: (1) the original dataset (labeled by GPT-4) as a baseline; (2) a Forced version where Llama-3B provides preference labels strictly based on numerical rewards; (3) a Filtered version where response pairs within the labeler's resolution are discarded; and (4) our Silent-aware approach (SymTie) which allows the labeler to stay silent and directly models these ties. The results reveal that simply *filtering out the identified ties leads to consistent improvements* for both the original and relabeled datasets, confirming that even golden benchmarks are contaminated by forced labels on indistinguishable samples. Furthermore, by modeling these ties, ***small and cost-effective models can yield alignment superior to*** that of more larger supervisors operating under the forced-choice paradigm.

While common strategies typically rely on larger models or refining prompt engineering to ensure feedback quality, our findings suggest that the forced-choice constraint itself is a fundamental bottleneck. By explicitly accommodating the labeler's discriminative resolution, we preserve the reliability of the alignment signal and enhance the overall training effectiveness. Specifically, we make the following key contributions:

**Contribution 1: A paradigm shift from forced choices to silent-aware framework.** We identify the limitations of the conventional binary forced-choice paradigm by observing the widespread presence of indistinguishable pairs across high-quality benchmarks. Inspired by psychophysics and semiorder theory, we introduce the discriminative resolution

*Table 1.* Alignment performance across different labelers and labeling paradigms on UltraFeedback. Performance is measured using the Gold Reward as a proxy for response quality.

| Annotator | Labeling Paradigm | Gold Reward |
|---|---|---|
| Golden (GPT-4) | Forced | 6.48 |
| Golden (GPT-4) | Filtered | 6.78 |
| Labeler (Llama-3B) | Forced | 7.94 |
| Labeler (Llama-3B) | Filtered | 8.06 |
| **Labeler (Llama-3B)** | **Silent-aware (SymTie)** | **8.20** |

limit to formally quantify the inherent cognitive boundaries of a labeler. Integrating this limit, we propose a labeling protocol that allows annotators to stay silent on ambiguous pairs, ensuring high-fidelity supervision. We demonstrate that *respecting the resolution limit matters more* for alignment than the model scale of the labeler. Specifically, an honest small labeler that remains silent on ambiguous pairs can outperform a forced larger counterpart or even human experts who may guess inconsistently, *unlocking the latent value* in existing benchmarks by reclaiming ties from pairs with near-identical scores previously obscured by forced preferences.

**Contribution 2: Extensive empirical validation of silent-aware optimization.** We propose several effective silent-aware objectives to optimize policy models using identified tie pairs while preventing arbitrary distribution shifts caused by forced-choice noise. Beyond identifying neutral ties, our framework refines these pairs into "Both-Good" and "Both-Bad" subsets using implicit reward means *at virtually no additional computational cost* — a dimension that remains unexplored in preference alignment. To leverage these, we propose objectives that effectively utilize these quality-attributed signals to drive the policy toward higher-reward regions. We further introduce a *corrected* risk estimator to mitigate potential overfitting caused by negative empirical risk. Through a comprehensive evaluation with various labeler configurations, we show that our silent-aware methods significantly outperform standard DPO and existing silent-aware baselines across both in-distribution and out-of-distribution tasks.

## 2. Preliminaries

### 2.1. Pairwise Preference and LLM Alignment

We define $\pi_\theta$ as an LLM policy parameterized by $\theta$, which maps a query sequence $x = (x_1, \ldots, x_m) \in \mathcal{X}$ to a probability distribution over the response sequences $y = (y_1, \ldots, y_n) \in \mathcal{Y}$. Typically, an LLM is first pretrained on a large, unlabeled text dataset, and then supervised fine-tuned on smaller, high-quality data to establish desired

abilities like dialogue through maximum likelihood estimation (Mishra et al., 2022; Ouyang et al., 2022). We denote the SFT-trained model as $\pi_{\text{ref}}$.

$\pi_{\text{ref}}$ is aligned with pairwise-preference data where pairs of responses are evaluated (Ziegler et al., 2019; Ramamurthy et al., 2023). For a prompt $x$, we denote $y_c \succ y_r$ if the response $y_c$ is preferred over $y_r$ by a labeler (e.g., human or a frontier LLM). An empirical pairwise-preference dataset $\mathcal{D} = \{(x^{(i)}, y_c^{(i)}, y_r^{(i)})\}_{i=1}^n$ consists of $n$ such triplets sampled from an underlying preference distribution $p^*$.

The preferences are assumed to be generated by some scalar-valued reward function $r^* : \mathcal{X} \times \mathcal{Y} \to \mathbb{R}$, which indicates the favorability of a response $y$ to the query $x$. Since the latent (real) reward function is usually not accessible, it is typically learned from pairwise preferences. The Bradley-Terry (BT) model provides the probabilistic foundation for modeling the probability distribution of pairwise comparison outcomes. It defines the probability that $y_c$ is ranked higher than $y_r$ as

$$P(y_c \succ y_r | x) = \frac{\exp(r^*(x, y_c))}{\exp(r^*(x, y_c)) + \exp(r^*(x, y_r))}. \quad (1)$$

Based on the BT model, a reward model $r$ learns to serve as a proxy of $r^*$, done by minimizing the negative log-likelihood:

$$\mathcal{R}(r) = -\mathbb{E}_{(x, y_c, y_r) \sim \mathcal{D}} \left[ \log \sigma \left( r(x, y_c) - r(x, y_r) \right) \right]. \quad (2)$$

Given the learned reward function, aligning the model is typically cast as a trade-off between maximizing expected reward and maintaining proximity to the SFT model $\pi_{\text{ref}}$ (also called reference model in alignment) in terms of KL-divergence:

$$\max_{\theta} \mathbb{E}_{\substack{x \sim \mathcal{D} \\ y \sim \pi_\theta(\cdot|x)}} [r(x, y)] - \beta D_{\text{KL}}(\pi_\theta(y|x) \| \pi_{\text{ref}}(y|x)), \quad (3)$$

where $\beta$ is a hyperparameter controlling the deviation from the SFT model. While reinforcement learning algorithms like PPO (Schulman et al., 2017) are commonly used to solve this problem, DPO proposes a more stable alternative without learning a separate reward model. By utilizing the closed-form solution of $r$, DPO optimizes the policy directly:

$$\mathcal{R}_{\text{DPO}}(\pi_\theta) = -\mathbb{E}_{(x, y_c, y_r) \sim \mathcal{D}}[\log \sigma(h_\theta(y_c, y_r; x))], \quad (4)$$

where $h_\theta(y_c, y_r; x) = \beta \log \frac{\pi_\theta(y_c|x)}{\pi_{\text{ref}}(y_c|x)} - \beta \log \frac{\pi_\theta(y_r|x)}{\pi_{\text{ref}}(y_r|x)}$.

### 2.2. Pointwise Supervision

An alternative approach to alignment is Kahneman-Tversky Optimization (KTO) (Ethayarajh et al., 2024), which shifts the focus from pairwise comparisons to point-wise utility. KTO uses binary feedback for individual responses,

where each $(x, y)$ is independently labeled as desirable ($y \in \mathcal{D}_{\text{pos}}$) or undesirable ($y \in \mathcal{D}_{\text{neg}}$). This format shifts the objective from margin-maximization to utility-optimization. The model is trained to maximize the utility $v_\theta(x, y) = \beta \log \frac{\pi_\theta(y|x)}{\pi_{\text{ref}}(y|x)}$ for positive samples and minimize it for negative ones relative to a reference point $z_{\text{ref}}$:

$$\mathcal{R}_{\text{KTO}}(\pi_\theta) = \mathbb{E}_{(x,y) \sim \mathcal{D}_{\text{pos}}} \left[ w_{\text{pos}} \left( 1 - \sigma \left( v_\theta(x, y) - z_{\text{ref}} \right) \right) \right]$$
$$+ \mathbb{E}_{(x,y) \sim \mathcal{D}_{\text{neg}}} \left[ w_{\text{neg}} \left( 1 - \sigma \left( z_{\text{ref}} - v_\theta(x, y) \right) \right) \right], \quad (5)$$

where $w_{pos}, w_{neg}$ are weights for the asymmetry in loss aversion.

## 3. Alignment with Silent-Aware Labeling

In this section, we introduce our silent-aware framework. We first formulate the discriminative resolution limit in Section 3.1, drawing on psychophysics and semiorder theory. Building on this, Section 3.2 details our labeling protocol for partitioning preference data and augmenting ties with quality tags. Sections 3.3 and 3.4 present our optimization objectives, ranging from neutral tie signals to quality-attributed ones, the latter of which is supported by a corrected risk estimator to ensure optimization stability.

### 3.1. The Discriminative Resolution Limit

To understand why the forced-choice paradigm may hinder alignment, we frame the labeling process through the lens of psychophysics, specifically the principle of the *just noticeable difference* (JND) (Davidson, 1970). JND explains why observers fail to reliably distinguish between two stimuli when the difference in their magnitude falls below a specific sensory threshold. This principle formalizes how humans and LLM-based labelers perceive quality variations in a resolution-limited manner: given the complexity of the response space, a labeler becomes insensitive to subtle quality difference. In LLM alignment, popular frameworks typically assume that a definitive preference exists for any two responses, thereby overlooking the de facto resolution limits of labelers. We argue that when response qualities are too similar, forcing a binary choice compels the labeler to provide unreliable preference. Consequently, we leverage the theory of *semiorders* (Luce, 1956), where a preference $x \succ y$ is only established if the latent quality difference exceeds a positive threshold $\delta_{\mathcal{M}}$. Semiorders provide a rigorous foundation for moving beyond binary constraints, naturally accommodating the inherent ties in human and LLM judgment.

Grounded in the semiorder framework, we posit that any labeler $\mathcal{M}$ (human or LLM) possesses a finite *discriminative resolution* $\delta_{\mathcal{M}} \geq 0$. A reliable preference $y_c \succ y_r$ can only be established if the latent reward difference $|r^*(x, y_c) - r^*(x, y_r)| > \delta_{\mathcal{M}}$. When the difference falls

within $[-\delta_{\mathcal{M}}, \delta_{\mathcal{M}}]$, the labeler reaches its resolution limit and cannot consistently distinguish the candidates. While forced binary choices pressure the labeler to rely on arbitrary heuristics or implicit biases in this region, our framework allows the labeler to stay silent.

**Definition 3.1** (Tied Preference Data). Consider two responses $y_1, y_2$ for an input prompt $x$. We denote $y_1 \approx y_2$ as a tied pair if the responses are deemed indistinguishable by the labeler $\mathcal{M}$ given its resolution limit $\delta_{\mathcal{M}}$. The empirical tied dataset $\mathcal{D}_{\text{tie}} = \{(x^{(i)}, y_1^{(i)}, y_2^{(i)})\}_{i=1}^{n_{\text{tie}}}$ consists of $n_{\text{tie}}$ such triplets where the quality difference is below the threshold:

$$\mathcal{D}^{\text{tie}} = \{(x, y_1, y_2) \mid |r_{\mathcal{M}}(x, y_1) - r_{\mathcal{M}}(x, y_2)| \le \delta_{\mathcal{M}}\}.$$

### 3.2. Silent-Aware Labeling Protocol

To equip the language model with the ability to provide preference feedback, we follow a protocol similar to (Tao & Li, 2025). Specifically, we begin with an initial SFT model $\pi_l^{\text{ref}}$ and train a policy model $\pi_l$ by optimizing the DPO objective (cf. Eq. (4)) on a labeled preference dataset $\mathcal{D}_{\text{pref}}$: $\pi_l = \arg\min_{\pi} \mathcal{R}_{\text{DPO}}(\pi_l; \pi_l^{\text{ref}}, \mathcal{D}_{\text{pref}})$. We use the subscript $l$ to indicate "labeler" in the remainder of the paper. For each triplet $(x, y_1, y_2)$, we compute the rewards $r_l(x, y_1)$ and $r_l(x, y_2)$ according to the DPO implicit reward formulation:

$$r_l(x, y) = \beta \log \frac{\pi_l(y|x)}{\pi_l^{\text{ref}}(y|x)}. \tag{6}$$

In a standard forced-choice paradigm, a preference label is assigned strictly based on the magnitude of these rewards:

$$\hat{y}_w = \begin{cases} y_1, & \text{if } r_l(x, y_1) > r_l(x, y_2), \\ y_2, & \text{otherwise.} \end{cases} \tag{7}$$

We denote the resulting labeled dataset as $\mathcal{D}_l^{\text{forced}} = \{(x, \hat{y}_w, \hat{y}_l)\}$.

To account for the labeler's resolution limit, we transition from binary labels to a threshold-based partitioning. We introduce a discriminative resolution parameter $\delta_{\mathcal{M}}$ that defines the boundary of indistinguishability following Def. 3.1. A response pair $(y_1, y_2)$ is categorized into the preferred subset $\mathcal{D}_l^{\text{pref}}$ according to Eq. (7) if the reward difference exceeds the threshold: $|r_l(x, y_1) - r_l(x, y_2)| > \delta_{\mathcal{M}}$. Conversely, pairs falling within this threshold, along with those where rewards are exactly equal, are categorized as the tied subset $\mathcal{D}_l^{\text{tie}} = \{(x, y_1, y_2)\}$, where $|\mathcal{D}| = |\mathcal{D}_l^{\text{forced}}| = |\mathcal{D}_l^{\text{pref}}| + |\mathcal{D}_l^{\text{tie}}|$.

While $\mathcal{D}_l^{\text{tie}}$ captures *neutral tie signals* between candidates, it does not reflect their quality. We can refine these tied pairs by estimating quality tags *at virtually no additional computational cost*. Specifically, we calculate the mean

implicit reward of the pair, $\bar{r} = (r_l(x, y_1) + r_l(x, y_2))/2$, and compare it to the global mean reward $\bar{r}_l^{\mathcal{D}}$ across all sampled responses in the dataset. A tied pair is augmented with a "Both-Good" tag to $\mathcal{D}_l^{\text{tie},+}$ if $\bar{r} > \bar{r}_l^{\mathcal{D}}$, and a "Both-Bad" tag to $\mathcal{D}_l^{\text{tie},-}$ otherwise.

### 3.3. Optimization with Neutral Tie Signals

**Optimization for Preferences.** For the preferred subset $\mathcal{D}_l^{\text{pref}}$, which comprises pairs where $y_c$ is robustly preferred over $y_r$, the alignment process can be facilitated by any objective designed for pairwise preference data, such as DPO or $\Psi$PO (Azar et al., 2024). Moreover, given that these pairs were explicitly filtered to ensure their reward difference exceeds the labeler's discriminative resolution $\delta_{\mathcal{M}}$ in our framework, we can leverage this prior to enforce a more conservative decision boundary. To this end, we augment the sigmoid term with a margin parameter $\delta$, formulating the loss as

$$\mathcal{L}_{\text{marginDPO}}(\pi_\theta) = - \left[\log \sigma \left(h_\theta(\hat{y}_w, \hat{y}_l; x) - \delta\right)\right]. \tag{8}$$

The $-\delta$ term effectively shifts the sigmoid's activation region, requiring the model to satisfy a minimum log-probability ratio $\beta \log \frac{\pi_\theta(\hat{y}_w|x)}{\pi_{\text{ref}}(\hat{y}_w|x)} - \beta \log \frac{\pi_\theta(\hat{y}_l|x)}{\pi_{\text{ref}}(\hat{y}_l|x)} > \delta$ before the optimization gradient diminishes. This ensures that the policy produces a quality gap consistent with the high-confidence signals provided by the labeler. This objective was also formally derived in Guo et al. (2025).

**Optimization for Ties.** For the tied subset $\mathcal{D}_l^{\text{tie}}$, we define the *symmetric tie loss*:

$$\begin{aligned} \mathcal{L}_{\text{symTie}}(\pi_\theta) = - \big[ &0.5 \log \sigma \left(h_\theta(y_1, y_2; x) - \delta\right) \\ &+ 0.5 \log \sigma \left(h_\theta(y_2, y_1; x) - \delta\right) \big] \end{aligned} \tag{9}$$

to utilize the neutral tie signals where the labeler remained silent. The gradient of the symmetric tie loss can be expressed as $\nabla_\theta \mathcal{L}_{\text{symTie}} = W(h_\theta) \cdot \nabla_\theta h_\theta$, where the weight function is $W(h_\theta) = 0.5 [\sigma(h_\theta - \delta) + \sigma(h_\theta + \delta) - 1]$. This weight function is monotonically increasing and satisfies $W(0) = 0$ due to the symmetry of the sigmoid function. When $h_\theta = 0$, the rewards of the two responses are identical and the gradient vanishes, maintaining relational consistency for indistinguishable pairs. By contrast, the gradient weight of standard DPO remains non-zero (0.5) even when rewards are identical. As a result, standard DPO continues to update policy models and potentially shifts the distribution toward a randomly chosen candidate. In contrast, $\mathcal{L}_{\text{symTie}}$ refrains from updating parameters once the reward margin approaches zero, thereby preserving a consistent neutral alignment.

The symmetric tie loss shares a mathematical connection with the classical Rao-Kupper model (Rao & Kupper, 1967), a well-established extension of the Bradley-Terry model

that explicitly accommodates ties. The Rao-Kupper model posits that a tie occurs when the latent reward difference $\mu = r^*(x, y_1) - r^*(x, y_2)$ is indistinguishable to the annotator, falling strictly within a discriminative threshold interval $[-\delta, \delta]$. Under this assumption, the probability of a tie is given by:

$$P(y_1 \approx y_2|x) = \frac{1}{1 + \exp(-\mu - \delta)} - \frac{1}{1 + \exp(-\mu + \delta)}. \tag{10}$$

By minimizing the negative log-likelihood of this tie probability, we obtain the loss function $\mathcal{L} = -\log(\exp(2\delta) - 1) + \log(1 + \exp(\mu + \delta)) + \log(1 + \exp(-\mu + \delta))$. In our framework, the resolution threshold $\delta$ is a predefined hyperparameter, meaning the first term acts as a constant that does not contribute to the optimization gradient. Dropping this constant and substituting the reward difference $\mu$ with the policy model's implicit reward margin $h_\theta(y_1, y_2; x)$, the optimization objective simplifies to:

$$\begin{aligned}\mathcal{L} &\propto \log(1 + \exp(h_\theta + \delta)) + \log(1 + \exp(-h_\theta + \delta)) \\ &= -\log \sigma(-h_\theta - \delta) - \log \sigma(h_\theta - \delta).\end{aligned} \tag{11}$$

Because $h_\theta(y_2, y_1; x) = -h_\theta(y_1, y_2; x)$ based on its definition, this equation perfectly recovers our proposed $\mathcal{L}_{symTie}$ (up to a scaling factor of 0.5).

### 3.4. Beyond Neutral Tie Signals

Given that quality tags are a cost-free byproduct of modern annotation pipelines — often derived from the same scoring functions (Guo et al., 2025) or human-judges (Zheng et al., 2023) — they are rich source of supervision. To the best of our knowledge, such information has remained entirely overlooked in existing alignment frameworks, where tied responses are typically discarded, or just treated with a blind symmetry (Guo et al., 2025). To leverage these *quality-attributed tie signals*, we propose two approaches.

**Deconstruct-then-KTO.** A direct approach to leverage quality tags is *deconstruct-then-KTO*. Following KTO, we transform tied pairs into point-wise supervision based on their identified quality. Specifically, responses in $\mathcal{D}_l^{\text{tie},+}$ are categorized as desirable outcomes, while those in $\mathcal{D}_l^{\text{tie},-}$ are categorized as undesirable. Specifically, for each $(x, y_1, y_2) \in \mathcal{D}_l^{\text{tie},+}$, we derive two independent desirable samples $(x, y_1)$ and $(x, y_2)$ to be reinforced. Conversely, for each pair in $\mathcal{D}_l^{\text{tie},-}$, we derive two undesirable samples $(x, y_1)$ and $(x, y_2)$ to be penalized. These are then optimized using the KTO objective (cf. Eq. (5)), which anchors the policy by increasing the reward of desirable outputs and decreasing it for undesirable ones relative to a learned reference point.

**Directional Tie Loss.** Alternatively, we propose the *directional tie loss*, which modulates the symmetric tie loss using

quality tags:

$$\mathcal{L}_{\text{dirTie}}(\pi_\theta) = \begin{cases} \mathcal{L}_{\text{symTie}}(\pi_\theta), & \text{if } (x, y_1, y_2) \in \mathcal{D}_l^{\text{tie},+}, \\ -\mathcal{L}_{\text{symTie}}(\pi_\theta), & \text{if } (x, y_1, y_2) \in \mathcal{D}_l^{\text{tie},-} \end{cases} \tag{12}$$

This loss introduces quality-attributed signals to alter the optimization landscape.

For Both-Good pairs, the objective reduces to $\mathcal{L}_{\text{symTie}}$, inheriting the stability properties established in the previous part. By exerting an *attractive force* that pulls the reward margin toward zero, the loss effectively acts as a relational anchor. In high-reward manifolds, this anchoring mechanism ensures the policy treats equally excellent candidates as equivalent, thereby preventing it from over-optimizing for negligible stylistic deltas and keeping the model's distribution focused on core quality rather than superficial variations.

For Both-Bad pairs, reversing the sign creates a *repulsive force* that prevents the rewards of both candidates from settling at an equal, low level. The second-order derivative of this directional loss is $\frac{\partial^2 \mathcal{L}_{\text{dirTie}}}{\partial h_\theta^2} = -0.5\big[\sigma(h_\theta - \delta)(1 - \sigma(h_\theta - \delta)) + \sigma(h_\theta + \delta)(1 - \sigma(h_\theta + \delta))\big]$. At $h_\theta = 0$, the curvature is $-\sigma(\delta)(1-\sigma(\delta))$, which turns the previous equilibrium into a *local maximum*. This ensures that any small update will push the policy away from these low-quality states, where the strength of this repulsion is controlled by the resolution threshold $\delta$. This mechanism is essential because, in standard symmetric optimization, a state where two poor responses have equal rewards is a local minimum. If the model produces two nonsensical answers, a standard tie loss would be perfectly satisfied, trapping the policy in a low-quality region. By making this equilibrium unstable, the directional loss pushes the model's probability mass away from these poor responses. Since the objective $\mathcal{L}_{\text{marginDPO}}$ simultaneously provides a clear gradient toward high-reward regions, this repulsive signal helps the policy "escape" low-quality states while still acknowledging that neither candidate in the bad pair is better than the other.

**Corrected Risk Estimator.** While the point-wise loss $\mathcal{L}_{\text{dirTie}}$ defines the local objective, in practice, we minimize the expected loss over the distribution of tied responses, which we formalize as the empirical risk:

$$\begin{aligned}\hat{\mathcal{R}}_{\text{dirTie}}(\pi_\theta) &= \mathbb{E}_{(x,y_1,y_2)\in\mathcal{D}_l^{\text{tie},+}} \mathcal{L}_{\text{symTie}}(\pi_\theta) \\ &- \mathbb{E}_{(x,y_1,y_2)\in\mathcal{D}_l^{\text{tie},-}} \mathcal{L}_{\text{symTie}}(\pi_\theta).\end{aligned} \tag{13}$$

However, this risk estimator incorporates negative sign for the Both-Bad pairs. Such estimators can be problematic: when training networks, the empirical risk could drop below zero. This negative empirical risk was observed be highly correlated with overfitting in recent studies (e.g., (Lu et al., 2020)), as the model exploits the negative components of the loss at the expense of generalization.

*Table 2.* Alignment performance on UltraFeedback with feedback from different labelers under varying resolution thresholds $\delta_{\mathcal{M}}$. The best results are highlighted in **bold**. Shaded columns represent methods utilizing neutral tie signals (pink) and quality-attributed tie signals (blue). Within each policy model and method, the optimal labeler and threshold configuration is marked with a dagger (†). Underlined values denote the DPO results obtained from the original dataset.

| Policy Model | Labeler | $\delta_{\mathcal{M}}$ | $n_{\text{tie}}$ | Forced | Filtered | TODO | SymTie | D-KTO | C-DirTie |
|---|---|---|---|---|---|---|---|---|---|
| Llama-3-8B | Llama-3B | 1 | 8,638 | 7.94 | 8.06 | 8.28 | 8.20 | 8.81 | **9.24** |
| | | 3 | 18,682 | | 8.29 | 8.57 | 8.39 | 9.92 | **10.28** |
| | | 5 | 28,043 | | 9.13† | 9.22 | 8.56 | 10.12 | **10.82** |
| | Llama-8B | 1 | 5,764 | 7.72 | 7.95 | 8.12 | 8.23 | 8.52 | **8.61** |
| | | 3 | 12,199 | | 8.18 | 8.99 | 8.89 | 9.77 | **10.23** |
| | | 5 | 18,771 | | 8.18 | 9.96† | 9.82† | 10.28† | **11.12**† |
| | **Golden (GPT-4)** | 0 | 10,368 | 6.48 | 6.78 | 8.82 | 8.48 | **9.10** | 8.81 |
| Mistral-7B | Llama-3B | 1 | 8,638 | 5.24 | 5.40 | 5.96 | 6.00 | 6.30 | **6.35** |
| | | 3 | 18,682 | | 6.27 | 6.55 | 6.23 | **6.88** | 6.76 |
| | | 5 | 28,043 | | 7.18† | 6.99† | 6.54† | **7.84**† | 7.32† |
| | Llama-8B | 1 | 5,764 | 5.33 | 5.06 | 5.14 | 5.11 | **5.78** | 5.25 |
| | | 3 | 12,199 | | 5.93 | 5.98 | 5.99 | 6.38 | **6.96** |
| | | 5 | 18,771 | | 6.27 | 6.68 | 6.42 | 6.78 | **6.97** |
| | **Golden (GPT-4)** | 0 | 10,368 | 4.40 | 5.13 | 5.76 | 6.00 | 6.54 | **6.67** |

To prevent this overfitting while still leveraging the quality tags, we propose a consistently corrected risk estimator

$$\hat{\mathcal{R}}_{\text{dirTie-cc}}(\pi_\theta) = f\left(\hat{\mathcal{R}}_{\text{dirTie}}(\pi_\theta)\right), \qquad (14)$$

where $f$ can be any functions that is non-negative, Lipschitz continuous, and satisfies $f(x) = x$ for all $x \geq 0$. Simple choices such as ReLU or absolute value functions are valid. In our experiments, we utilize the generalized leaky ReLU, i.e., $f(x) = \mathbb{I}_{x \geq 0} x + \mathbb{I}_{x < 0} \lambda x$.

**Unified Optimization Objective.** We formulate the optimization task as an empirical risk minimization (ERM) problem. The final objective is to minimize the total empirical risk defined as the weighted combination of the preference alignment risk and the risk derived from tied pairs:

$$\hat{\mathcal{R}}_{\text{total}} = \hat{\mathcal{R}}_{\text{pref}} + \alpha \hat{\mathcal{R}}_{\text{silent}}. \qquad (15)$$

where $\hat{\mathcal{R}}_{\text{pref}} = \mathbb{E}_{(x, y_1, y_2) \in \mathcal{D}_{\text{pref}}} \mathcal{L}_{\text{marginDPO}}(\pi_\theta)$, and $\hat{\mathcal{R}}_{\text{silent}}$ is the standard empirical risk derived from the tied subset. For brevity, the methods based on the losses $\mathcal{L}_{\text{symTie}}$ and $\mathcal{L}_{\text{dirTie}}$, and the risk estimator $\hat{\mathcal{R}}_{\text{dirTie-cc}}$ are respectively denoted as **SymTie**, **D-KTO**, and **C-DirTie**.

## 4. Silent-Aware Framework is A Feasible and Cost-Effective Trajectory for Alignment

### 4.1. Experimental Setups

**Models.** We utilize two representative open-sourced architectures, Mistral-7B-v0.1 (Jiang et al., 2023) and Llama-3-8B (Team, 2024). To provide a standardized baseline for instruction-following, we utilize the SFT versions of these models as our starting policies $\pi_{\text{ref}}$.

**Datasets.** We conduct our experiments on two large-scale preference datasets: UltraFeedback and ChatArena (Zheng et al., 2023). We divide the original training data into two disjoint subsets with a 1:7 ratio following Tao & Li (2025). The first subset is used to train a proxy labeler $\pi_l$ by optimizing the DPO objective. This labeler is then employed to generate preference annotations for the remaining subset, which subsequently serves as the training set for our alignment experiments.

To simulate labelers with varying levels of discriminative ability, we employ a hierarchy of LLMs, Llama-3B, Llama-8B, and Qwen-1.7B, Qwen-4B (Yang et al., 2025a), serving as the policy $\pi_l$ to derive implicit rewards following Eq. (6). Following the protocol in Section 3.2, the reward margins computed from these proxy labelers are partitioned across three discriminative resolutions $\delta_{\mathcal{M}} \in \{1, 3, 5\}$ to generate ties. By treating response pairs whose reward margins fall within the indistinguishable range $[-\delta_{\mathcal{M}}, \delta_{\mathcal{M}}]$ as ties, we examine how alignment performance fluctuates when a labeler's resolution limit is explicitly acknowledged.

In addition, we extract ties directly from the original datasets to serve as our golden baseline. For UltraFeedback, golden ties are response pairs with strictly identical GPT-4 scores, with quality-attributed signals assigned following the mean-reward protocol in Section 3.2. For ChatArena, we leverage the crowdsourced human annotations, which inherently distinguish between Both-Good and Both-Bad ties.

**Baselines.** To evaluate the efficacy of our approach, we compare our proposed objectives against three approaches that can handle tie samples:

- Forced: Preferences are strictly assigned according to the

*Table 3.* Alignment performance Llama-3-8B on ChatArena with feedback from different labelers.

| Policy Model | Labeler | $\delta_{\mathcal{M}}$ | $n_{\text{tie}}$ | Forced | Filtered | TODO | SymTie | D-KTO | C-DirTie |
|---|---|---|---|---|---|---|---|---|---|
| Mistral-7B | Llama-3B | 1 | 3,314 | 7.59 | 7.69 | 7.97 | 7.91 | 8.30 | **8.52** |
| | | 3 | 7,901 | | 7.76 | 8.44 | 8.49 | 8.96 | **9.34** |
| | | 5 | 11,759 | | 8.09$^\dagger$ | 9.13 | 9.20$^\dagger$ | 9.50 | **9.97** |
| | Llama-8B | 1 | 1,939 | 7.43 | 7.59 | 7.88 | 7.74 | 8.33 | **8.69** |
| | | 3 | 4,845 | | 7.48 | 8.69 | 8.47 | 9.41 | **9.84** |
| | | 5 | 7,463 | | 7.91 | 9.54$^\dagger$ | 8.75 | 10.06$^\dagger$ | **10.64**$^\dagger$ |
| | **Golden (Human)** | 0 | 7,957 | 6.73 | 7.29 | 8.80 | 8.51 | 9.71 | **9.99** |
| Mistral-7B | Llama-3B | 1 | 3,314 | 4.92 | 5.65 | 5.36$^\dagger$ | 5.58$^\dagger$ | **6.34** | 5.45 |
| | | 3 | 7,901 | | 6.32 | 5.51 | 5.26 | **6.54** | 6.06 |
| | | 5 | 11,759 | | 7.15$^\dagger$ | 5.06 | 4.98 | **7.29**$^\dagger$ | 6.52 |
| | Llama-8B | 1 | 1,939 | 4.86 | 5.23 | 5.34 | 5.25 | 5.85 | **5.90** |
| | | 3 | 4,845 | | 5.85 | 5.13 | 4.95 | 6.41 | **6.81** |
| | | 5 | 7,463 | | 5.67 | 5.09 | 5.46 | **7.14** | 6.93$^\dagger$ |
| | **Golden (Human)** | 0 | 7,957 | 3.41 | 4.42 | 4.07 | 3.33 | **4.80** | 4.72 |

numerical rewards following Eq. (7), where the response with the higher score is designated as the winner regardless of the margin's magnitude. Random tie-breaking via a coin-flip is invoked in cases of absolute score equality. This setup simulates the conventional forced-choice paradigm standard in existing benchmarks, which compels a definitive judgment despite stimuli approaching or falling below their discriminative threshold.

- Filtered: All response pairs with a reward margin smaller than the discriminative threshold $\delta_{\mathcal{M}}$ are discarded from the training set, that is, the model is aligned exclusively on the high-confidence preferred subset $\mathcal{D}_l^{\text{pref}}$.
- TODO (Guo et al., 2025): A state-of-the-art algorithm extending the BT model that explicitly incorporates ties for handling tie data in preference learning.

**Evaluation Metrics.** We evaluate generative performance using a held-out test set of 2,000 samples and the well-established benchmarks:

- Gold reward: Following (Gao et al., 2023; Coste et al., 2024; Tao & Li, 2025), we utilize the gold reward as an objective proxy for response quality. Due to the cost constraints of human evaluation, we employ the output of Skywork/Skywork-Reward-V2-Llama-3.1-8B (Liu et al., 2025a), a top-performing model on RewardBench leaderboard (Malik et al., 2025), to score model generations. A higher gold reward indicates better alignment.
- AlpacaEval (Li et al., 2023): To evaluate instruction-following in an out-of-distribution setting, we report the (length-controlled) win rates (Dubois et al., 2024) against the text_davinci_003 evaluated by GPT-4.
- Zero-shot/Few-shot Accuracy: We evaluate models on a suite of downstream reasoning and knowledge benchmarks, including MT Bench (Zheng et al., 2023) PIQA (Bisk et al., 2020), ARC-e (Clark et al., 2018), Hel-

laSwag (Zellers et al., 2019), MMLU (Hendrycks et al., 2021), and Winogrande (Sakaguchi et al., 2020).

All experiments were conducted across 3 trials, and we report the mean performance for all metrics. For a comprehensive description of the hyperparameters employed in our experiments, please refer to Appendix A.

### 4.2. Main Results

In this section, we present the primary findings, focusing on how different paradigms for handling indistinguishable pairs and how different silent-aware optimizations affect alignment quality. In Tables 2 and 3, we evaluate the gold reward of policies trained on feedback from labelers with varying discriminative resolution limits.

**Forced Choices Introduce Noise.** A consistent trend is that Filtered outperforms Forced. As shown in the "Golden" row, even GPT-4 and human labels are compromised by forced-choice constraints. Simply excluding pairs that the labeler cannot reliably distinguish increases the gold reward from 6.48 to 6.78 for Llama-3-8B and from 4.40 to 5.13 for Mistral-7B on UltraFeedback. This confirms that conventional forced-choice paradigm compels labelers to provide unreliable judgments on samples falling below their discriminative resolution, ultimately introducing noise that degrades policy optimization.

**Tie Signals as Powerful Regularization.** Under the traditional forced-choice paradigm, feedback from task-specific smaller models achieves better performance than that of frontier models or humans. This aligns with the main observation in Tao & Li (2025), suggesting that weak models feedback is a promising path for alignment. Crucially, our study reveals that the silent-aware framework significantly amplifies this cost-effectiveness of this path. While filtering

*Table 4.* AlpacaEval win rates of Llama-3-8B aligned with different labelers on UltraFeedback.

| Labeler | $\delta_{\mathcal{M}}$ | Forced | Filtered | TODO | SymTie | D-KTO | C-DirTie |
|---|---|---|---|---|---|---|---|
| | 1 | | 41.63 | 41.54 | 42.30 | 42.09 | **43.32** |
| Llama3B | 3 | 42.77 | **44.23**[†] | 44.18 | 41.92 | 44.18 | 43.22 |
| | 5 | | 42.47 | 43.88[†] | 42.25 | **46.62**[†] | 44.20 |
| | 1 | | 40.31 | 41.99 | 41.18 | **42.49** | 42.12 |
| Llama8B | 3 | 43.31 | 42.86 | 43.76 | 43.64 | 44.60 | **45.27** |
| | 5 | | 43.23 | 43.10 | 43.55 | **46.15** | 45.96[†] |
| **Golden** | 0 | 37.51 | 42.44 | 43.02 | 44.07[†] | 43.52 | 44.50 |

*Table 5.* MT Bench results of Llama-3-8B aligned with different labelers on UltraFeedback.

| Labeler | $\delta_{\mathcal{M}}$ | Forced | Filtered | TODO | SymTie | D-KTO | C-DirTie |
|---|---|---|---|---|---|---|---|
| | 1 | | 6.20 | 6.16 | 6.17 | **6.23** | 6.17 |
| Llama3B | 3 | 6.03 | 6.31[†] | 6.25[†] | 6.14 | 6.39 | **6.51**[†] |
| | 5 | | 6.14 | 6.21 | 6.11 | **6.54**[†] | 6.42 |
| | 1 | | 6.13 | 6.03 | 5.94 | 6.18 | **6.33** |
| Llama8B | 3 | 6.08 | 6.17 | 6.18 | 6.10 | **6.24** | 6.23 |
| | 5 | | 6.14 | 6.03 | 6.16 | 6.32 | **6.41** |
| **Golden** | 0 | 5.69 | 5.84 | 6.06 | 6.26[†] | 6.32 | **6.33** |

improves performance by removing ambiguous samples, all methods that directly leverage tie signals consistently surpass the Filtered baseline. This suggests that tie data is not merely noise to be discarded; instead, it serves as a form of regularization. By ensuring the gradient diminishes as the reward margin approaches the resolution limit, modeling ties flattens the optimization landscape in ambiguous regions and prevents arbitrary distribution shifts.

**Honesty Outweighs Labeler Scale.** As expected, at the same threshold $\delta_{\mathcal{M}}$, smaller models generate a higher number of ties ($n_{\text{tie}}$), reflecting their lower discriminative resolution. Interestingly, as $\delta_{\mathcal{M}}$ increases — effectively expanding the "zones of silence" — the alignment performance across all silent-aware methods improves consistently. This indicates that signal purity is far more critical than signal quantity. Even when more than half of the samples are treated as ties by 3B model at $\delta_{\mathcal{M}} = 5$, the policy benefits from the higher-quality gradients of confident samples. Second, we can observe that while the 8B labeler provides slightly higher peak performance than the 3B labeler, but the gap is remarkably small compared to the leap from forced-choice to silent-aware labeling. This is most evident when comparing Llama-8B at $\delta_{\mathcal{M}} = 1$ with Llama-3B at $\delta_{\mathcal{M}} = 5$: forcing the 8B model to judge nearly 90% of pairs introduces more harm than the 3B model being highly conservative. Consequently, the more honest 3B model achieves higher gold rewards than the 8B model at a lower threshold. Respecting a model's resolution limit is thus a highly cost-effective strategy for scaling supervision.

*Table 6.* Results of Llama-3-8B aligned with different methods on UltraFeedback. Feedback is provided by a Llama-3B labeler with $\delta_{\mathcal{M}} = 3$.

| Method | Piqa | ARC-e | MMLU | Hellaswag | Winogrande | Average |
|---|---|---|---|---|---|---|
| **Forced** | 79.02 | 74.78 | 63.89 | 76.28 | 67.32 | 72.26 |
| **Filtered** | 79.60 | 74.43 | 63.68 | 76.68 | 67.72 | 72.42 |
| **TODO** | 79.65 | 75.31 | 63.75 | **76.79** | 67.80 | 72.66 |
| **SymTie** | 79.87 | **75.49** | 63.46 | 76.54 | 68.19 | 72.71 |
| **D-KTO** | 80.25 | 74.25 | 63.83 | **76.79** | 68.51 | 72.72 |
| **C-DirTie** | **80.36** | 75.19 | **63.93** | 76.58 | **68.59** | **72.93** |

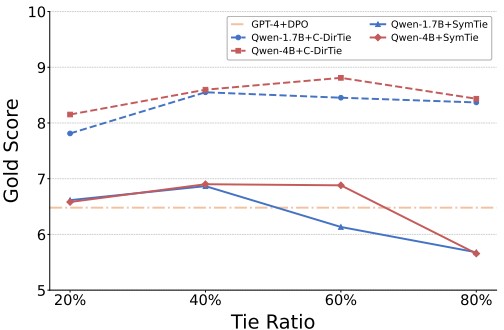

*Figure 2.* Alignment performance of Llama-3-8B using feedback from the Qwen model family.

**Quality-Attributed Signals Drive Maximum Gains.** With neutral tie signals, our method SymTie achieves performance comparable to the SOTA method TODO. And the most substantial gains are observed when introducing quality-attributed tie signals. By incorporating these easily accessible signals, our proposed C-DirTie and D-KTO generally achieve the highest rewards. While D-KTO exhibits superior peak performance on Mistral-7B, C-DirTie tends to dominate Llama-3-8B evaluations. This may suggest that different models possess varying sensitivities to absolute quality signals (leveraged by D-KTO) versus repulsive directional forces (leveraged by C-DirTie).

**Unlocking Latent Value in Golden Benchmarks.** The "Golden" row utilizes original labels without any additional re-labeling. By simply re-modeling the existing UltraFeedback data through our objectives, we achieve a substantial leap in gold reward; for instance, from $7.48$ to $9.10$ for the Llama-3-8B policy. By reclaiming ties from pairs with near-identical scores previously obscured by forced preferences, we successfully unlock the latent value in benchmarks that has been underestimated. This further suggests that the performance of current benchmarks is limited not only by the inherent quality of annotators judgments, but also by rigid forced-choice learning paradigms that fail to respect the supervisor's discriminative resolution limit. By allowing the model to stay silent on these near-identical pairs, our framework effectively purifies the supervision, thereby

facilitating a more robust and efficient alignment process.

**Robustness Across Policy Models and Datasets.** We evaluate our approach on the Mistral-7B policy and the ChatArena dataset to verify the consistency of our findings. As shown in Tables 2 and 3, the results follow the same patterns observed with Llama-3-8B on UltraFeedback. Notably, the sustained gains on ChatArena, which relies on human annotators, demonstrate that our silent-aware framework effectively leverages the resolution limits of both human and AI labelers. This consistency across different policy architectures and feedback sources underscores that the benefits of our framework are source-agnostic and not over-fitted to a specific experimental setup.

### 4.3. Ablation Studies

**Generalization Across Downstream Benchmarks.** To ensure that the observed improvements are not overfitted to a specific reward model, we evaluate our policies on AlpacaEval using GPT-4 as the automatic judge. As shown in Table 4, the win rates trajectories closely mirror the Gold Reward results presented in Section 4.2. We further assess the performance of our methods across a series of standard benchmarks to evaluate general reasoning and knowledge preservation. As demonstrated in Table 6, our approaches enhance performance across diverse tasks compared with Forced and Filtered baselines. This cross-metric consistency confirms that the benefits of the silent-aware framework translate effectively from static reward scores to complex downstream tasks.

**Results on Different Model Families.** To verify that our findings are architecture-agnostic, we employ the Qwen series models as proxy labelers to provide feedback for the Llama-3-8B policy model using SymTie and C-DirTie. we define ties by categorizing the pairs with the lowest $20\%$, $40\%$, $60\%$, and $80\%$ score differences as indistinguishable. This percentile-based approach allows us to observe the scaling behavior of silent-aware optimization under varying degrees of "conservativeness" regardless of the labeler's raw reward scale. As shown in Figure 2, the alignment performance exhibits a consistent inverted U-shape across all configurations. As the tie ratio increases initially, performance rises and surpasses the golden baseline (6.48), confirming that the effectiveness of our silent-aware framework holds true across diverse model families. And the performance begins to decline when the tie ratio exceeds a critical threshold, indicating a trade-off where excessive data discarding leads to insufficient supervision.

## 5. Discussion and Conclusion

In this work, we present a silent-aware framework that shifts from forced choices to a paradigm respecting the labeler's

discriminative resolution limit. Our findings reveal that the framework is *not only a labeling alternative but a highly feasible and cost-effective trajectory for robust RLAIF*. Further discussions regarding the connection between our findings and related works are deferred to Appendix B.

## Acknowledge

This research was supported by the Jiangsu Science Foundation (BG2024036, BK20243012, BK20241297), the National Science Foundation of China (62406066, 62125602, 62576093, U24A20324, 92464301), the New Cornerstone Science Foundation through the XPLORER PRIZE, and the Fundamental Research Funds for the Central Universities (2242025K30024), and the Big Data Computing Center of Southeast University.

## Impact Statement

This paper presents work whose goal is to advance the field of Machine Learning. There are many potential societal consequences of our work, none which we feel must be specifically highlighted here.

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

# A. Experiments

## A.1. Training Details

**Labeler.** For the Llama series, we adopt their respective official instruct versions Llama-3.2-3B-Instruct (Liu et al., 2025b) and Llama-3.1-8B-Instruct (Patterson et al., 2022) as the reference policies $\pi_l^{\text{ref}}$. For the Qwen series, we employ the instruct versions of Qwen3-1.7B and Qwen3-4B (Yang et al., 2025a) as the reference policies $\pi_l^{\text{ref}}$. All models undergo DPO fine-tuning using a randomly sampled $1/8$ subset of the training data. For the DPO training phase, we perform full-parameter fine-tuning on all models. We set the number of epochs to 12, the learning rate to $5 \times 10^{-6}$, and the DPO divergence parameter $\beta$ to 0.1. The batch size is fixed at 8 with a gradient accumulation step of 2. We set the maximum sequence length to 1024 tokens. all parameters

**Datasets.** As illustrated in Figure 1, our analysis incorporates results from four distinct benchmarks. Beyond the primary datasets, we include HelpSteer2 (Wang et al., 2025), whose golden labelers for this dataset are human annotators. Following the official recommendation, we identify a tie when the preference strength is explicitly labeled as 0. Additionally, we utilize Orca-dpo-pairs (Mukherjee et al., 2023), which utilizes GPT-4 as the golden labeler. Ties are defined as instances where GPT-4 assigns identical scores to both response candidates. For the four datasets showcased in Figure 1, the scale parameters $\theta$ are set to 3, 6, 3, and 20, respectively.

**Policy Model.** To ensure a fair comparison, we maintain consistent training hyperparameters across all evaluated methods, datasets, and labelers. We set the number of epochs to 1, the learning rate to $5 \times 10^{-6}$, and the DPO divergence parameter $\beta$ to 0.1. The batch size is fixed at 8 with a gradient accumulation step of 2. We set the maximum sequence length to 1024 tokens. For our proposed methods, we set $\alpha = 1$ in Eq. (15) and $\lambda = 0.75$ for leaky-ReLU in C-DirTie. Regarding the margin parameter $\delta$, we consistently set its value to half of the resolution threshold $\delta_{\mathcal{M}}$ across all experiments, acting as a conservative boundary for confident preference.

**Hardware.** Our experiments are conducted on servers equipped with NVIDIA H100 GPUs, with 80 GB of VRAM.

## A.2. Silent-Aware Labeling

To operationalize the transition from traditional forced-choice paradigms to our silent-aware framework, we detail the structured process for synthesizing labelers and the whole labeling protocol. The workflow is formalized in Algorithm 1.

## A.3. Additional Analysis

**Qualitative Examples.** To intuitively demonstrate the efficacy of the silent-aware principle, we analyze qualitative examples from $\mathcal{D}_{\text{tie}}$ generated by our Llama-3B labeler with a resolution limit of $\delta = 3$. As illustrated in Table 7, In these instances, the golden labeler GPT-4 subjected to the conventional forced-choice constraint, was required to designate a winner, whereas our framework permitted the supervisor to remain silent. As illustrated in Table 7, although the golden labeler assigned definitive preferences, the actual quality of the responses remains remarkably similar.

**Intrinsic Consistency of the Resolution Limit Across Labelers.** We analyze the overlap of ties between different labelers in Table 8. As shown in the statistics, while the smaller Llama-3B consistently identifies a larger volume of ties than Llama-8B at the same threshold $\delta_{\mathcal{M}}$, their "zones of silence" exhibit high congruence. Specifically, even at a stringent resolution limit of $\delta_{\mathcal{M}} = 1$, the overlap ratio reaches 43.1%, accounting for nearly half of the ties identified by the more discriminative Llama-8B model. This overlap ratio $R_{8 \to 3}$ increases significantly as the threshold widens, reaching 74.5% at $\delta_{\mathcal{M}} = 5$. This suggests that the tie set of the larger, more discriminative model is largely a subset of the smaller model's tie set. This finding supports our hypothesis that the discriminative resolution limit is a fundamental property of the model-data interaction: as model scale increases, the region where the model lacks sufficient resolution to distinguish samples shrinks, yet the core area of indistinguishable samples remains remarkably consistent across scales. To further eliminate the influence of score magnitude across different architectures, we define ties using the bottom $K\%$ of score differences. As shown in Table 9, the overlap between Qwen-1.7B and Qwen-4B remains consistently high and stable across both $R_{1.7 \to 4}$ and $R_{4 \to 1.7}$, even at narrow thresholds (e.g., 45.6% and 41.2% at $K = 20\%$).

**Applicability to Alternative Preference Alignment Algorithms** We extend our evaluation beyond standard DPO to other recent preference alignment approaches, specifically CPO (Guo et al., 2024b) and SimPO (Meng et al., 2024). Table 10 presents the performance on UltraFeedback using the golden labeler (GPT-4) and a weak proxy labeler (Llama-8B) with a resolution threshold of $\delta_{\mathcal{M}} = 3$. The results indicate that replacing the conventional forced-choice paradigm with our

*Table 7.* Qualitative examples in $\mathcal{D}_{\text{tie}}$. Left: Both-Good ties where both responses are high-quality. Right: Both-Bad ties where both are low-quality. Preference labels and reward scores are the original values assigned by GPT-4 in UltraFeedback.

**Example 1**
**Prompt:** What is the largest bird in North America?
**Chosen:** The largest bird in North America is the California condor (Gymnogyps californianus). Adult California condors have a wingspan of up to 9.5 feet (2.9 meters) and can weigh up to 26 pounds (12 kilograms). (Reward GPT-4: **9.0**)
**Rejected:** The largest bird in North America is the California Condor (Gymnogyps californianus). With a wingspan of up to 9.5 feet (2.9 meters) and a weight of up to 26 pounds (12 kilograms), it is considered the largest flying bird on the continent. This critically endangered species is a member of the New World vulture family and primarily feeds on carrion. (Reward: **9.0**)
*GPT-4 forced chosen over rejected, but both are excellent.*

**Example 3**
**Prompt:** Definition: Given a sentence in Spanish, provide an equivalent paraphrased translation in French that retains the same meaning both through the translation and the paraphrase.
Input: Ahora resuelva el precio de oferta de indiferencia para la Fórmula 31 para resolver.
Output:
**Chosen:** Resuelva ahora la fórmula de descuento para la Fórmula 31. (Reward: **2.0**)
**Rejected:** Provide the price of indifference offer for Formula 31 to solve. (Reward: **2.0**)
*Both options fail to meet the requirements; neither should be chosen.*

**Example 2**
**Prompt:** What are some health benefits of consuming dark chocolate?
**Chosen:** Dark chocolate, especially when containing at least 70% cocoa, is not only a delicious treat but also comes with a variety of health benefits. Its consumption can spark curiosity in the fields of nutrition and health, as it offers a blend of taste and well-being. Here are some noteworthy health benefits of consuming dark chocolate:... So next time you're tempted to have a treat, consider satisfying your sweet tooth with some delicious and healthful dark chocolate! (Reward: **8.5**)
**Rejected:** Dark chocolate, particularly those with high cocoa content (70% or higher), is known to have several health benefits due to its rich source of antioxidants and nutrients. Some of these benefits include:... As with any food, it's essential to maintain balance and moderation for optimal health. (Reward: **8.5**)
*Minimal margin between two responses makes preference label stochastic.*

**Example 4**
**Prompt:** Which one exists in human cells, cytoderm or chloroplast?
**Chosen:** Chloroplasts exist in human cells. In addition to providing energy in the form of glucose through photosynthesis in plant cells, chloroplasts also play a significant role in other cellular processes. They are found in small quantities in some human cell types, such as those in the gastrointestinal and female reproductive systems. These chloroplasts are thought to play a role in processing vitamins and antioxidants, but they don't contribute to energy production in the way they do in plant cells. So, while there are chloroplasts in human cells, there are no cytoderms in any cells. (Reward: **3.0**)
**Rejected:** Chloroplasts exist in human cells. (Reward: **3.0**)
*Both options are flawed; neither should be chosen.*

silent-aware objectives consistently yields significant performance gains across all evaluated baseline algorithms. This confirms that the regularization benefits of respecting a labeler's resolution limit are orthogonal to the specific pairwise optimization objective employed.

**Robustness Under Varying Label Noise** To further validate the robustness of our labeling pipeline, we investigated the impact of varying tie ratios during the proxy labeler's training phase. We synthesized training sets with controlled noise levels by selecting 1/8 of the total data as a clean preference dataset from UltraFeedback (where the definitive GPT-4 score gap is $\geq 3$). We then randomly replaced $20\%$ and $40\%$ of this subset with "tie data" (pairs with identical GPT-4 scores) but assigned them forced, random binary preference labels. A proxy labeler was trained on each set and subsequently used to annotate the full dataset using a uniform threshold of $\delta_{\mathcal{M}} = 3$.

As shown in Table 11, policy models aligned using the noise-injected labelers ($20\%$ tie ratio) surprisingly outperformed the clean baseline ($0\%$ tie ratio). This phenomenon may be closely tied to the capacity constraints of smaller models. When a labeler is trained exclusively on easily distinguishable pairs, it tends to become overconfident and rarely outputs small reward margins. Conversely, exposing the model to forced-choice noise on indistinguishable pairs provides contradictory signals that a capacity-limited neural network struggles to memorize (Arpit et al., 2017). Consequently, the model naturally remains unconfident on ambiguous pairs, yielding smaller reward margins. This finding demonstrates that applying a hard

---

**Algorithm 1** Silent-Aware Labeling.

---

1: **Input:** Initial SFT model $\pi_l^{\text{ref}}$, labeled preference dataset $\mathcal{D}_0^{\text{pref}}$, unlabeled triplets $\mathcal{D}_u = \{(x, y_1, y_2)\}$, resolution limit $\delta_{\mathcal{M}}$, and reward coefficient $\beta$.

2: **Phase 1: Labeler Synthesis**

3: Train labeler $\pi_l \leftarrow \arg\min_\pi \mathcal{L}_{\text{DPO}}(\pi; \pi_l^{\text{ref}}, \mathcal{D}_0^{\text{pref}})$

4: **Phase 2: Implicit Reward Scoring**

5: **for** each $(x, y_1, y_2) \in \mathcal{D}_u$ **do**

6:     $r_l(x, y_1) \leftarrow \beta \log \frac{\pi_l(y_1|x)}{\pi_l^{\text{ref}}(y_1|x)}$

7:     $r_l(x, y_2) \leftarrow \beta \log \frac{\pi_l(y_2|x)}{\pi_l^{\text{ref}}(y_2|x)}$

8: **end for**

9: Calculate global mean reward $\bar{r}_l^{\mathcal{D}} \leftarrow \text{mean}(\{r_l(x, y) \mid y \in \mathcal{D}_u\})$

10: **Phase 3: Silent-Aware Partitioning**

11: Initialize $\mathcal{D}_l^{\text{pref}} \leftarrow \emptyset, \mathcal{D}_l^{\text{tie},+} \leftarrow \emptyset, \mathcal{D}_l^{\text{tie},-} \leftarrow \emptyset$

12: **for** each $(x, y_1, y_2) \in \mathcal{D}_u$ **do**

13:     $\Delta r \leftarrow |r_l(x, y_1) - r_l(x, y_2)|$

14:     **if** $\Delta r > \delta_{\mathcal{M}}$ **then**

15:         Determine $\hat{y}_w, \hat{y}_l$ via Eq. (7) and add to $\mathcal{D}_l^{\text{pref}}$

16:     **else**

17:         $\bar{r} \leftarrow (r_l(x, y_1) + r_l(x, y_2))/2$

18:         **if** $\bar{r} > \bar{r}_l^{\mathcal{D}}$ **then**

19:             Add $(x, y_1, y_2)$ to $\mathcal{D}_l^{\text{tie},+}$ {*Both-Good*}

20:         **else**

21:             Add $(x, y_1, y_2)$ to $\mathcal{D}_l^{\text{tie},-}$ {*Both-Bad*}

22:         **end if**

23:     **end if**

24: **end for**

25: **Output:** Partitioned datasets $\mathcal{D}_l^{\text{pref}}$ and $\mathcal{D}_l^{\text{tie}} = \mathcal{D}_l^{\text{tie},+} \cup \mathcal{D}_l^{\text{tie},-}$

---

margin threshold $\delta_{\mathcal{M}}$ is not merely an empirical compromise, but rather a highly reliable and computationally free proxy for capturing the model's latent epistemic uncertainty.

**Integration with EM-style Tie Estimation** Our primary pipeline relies on a static preprocessing step, using reward margins from a separately trained proxy labeler to define the "zones of silence" before training begins. Inferring sample uncertainty via Expectation-Maximization (EM) style estimation can offer an alternative. To demonstrate the compatibility of our silent-aware framework with dynamic uncertainty estimation, we adapted RE-PO (Cao et al., 2026) to our setting. Specifically, instead of identifying ties offline at the very beginning, we dynamically identify them on-the-fly at each training step. During the E-step at iteration $t$, RE-PO computes a posterior confidence score $w_i^{(t)}$ for each sample $i$, representing the probability that the annotated preference is correct given the current policy $\theta^{(t)}$ and the estimated annotator reliability. A score $w_i^{(t)}$ approaching 0.5 indicates maximum epistemic uncertainty, implying that the two responses are virtually indistinguishable under the current model and annotator prior. We leverage this dynamic metric to redefine our "zones of silence": within each mini-batch, if a sample's posterior probability $w_i^{(t)}$ falls into a predefined highly uncertain confidence interval (e.g., $[0.45, 0.55]$ or $[0.40, 0.60]$), it is classified as a tie for that specific update step and optimized using our symmetric tie loss. Samples falling outside these intervals are treated as definitive preferences and processed accordingly.

Table 12 reports the performance across DPO, CPO, and SimPO. Even within an EM-style framework, explicitly modeling the identified ties via SymTie consistently outperforms both the standard RE-PO and the Filtered approach. This further corroborates our core hypothesis: properly modeling tie signals prevents arbitrary distribution shifts and provides crucial optimization stability, regardless of whether the ambiguity is identified via static margin thresholds or dynamic posterior probabilities.

*Table 8.* Tie overlap between Llama-3B and Llama-8B labelers. $R_{8 \to 3}$ denotes the percentage of ties found by the 3B model that were also identified by the 8B model.

| Threshold $\delta_{\mathcal{M}}$ | Ties (8B) | Ties (3B) | $R_{3 \to 8}$ |
|---|---|---|---|
| 1 | 5,764 | 8,638 | 43.1% |
| 3 | 12,199 | 18,682 | 59.0% |
| 5 | 18,771 | 28,043 | 74.5% |

*Table 9.* Tie overlap ratio between Qwen-1.7B and Qwen-4B labelers using percentage-based thresholds ($K\%$). $R_{4 \to 1.7}$ represents the intersection over the smaller model's tie set and $R_{1.7 \to 4}$ inversely.

| Bottom $K\%$ | 20% | 40% | 60% | 80% |
|---|---|---|---|---|
| $R_{1.7 \to 4}$ | 45.6% | 62.8% | 79.4% | 90.5% |
| $R_{4 \to 1.7}$ | 41.2% | 63.4% | 82.1% | 89.5% |

# B. Related Works and Discussions

## B.1. Foundations of LLM Alignment

Aligning large language models (LLMs) (Christiano et al., 2017; Ouyang et al., 2022; Bai et al., 2022a; Xia et al., 2024a;b) with human intent is fundamental to ensuring their safety and utility. The standard reinforcement learning from human feedback (RLHF) framework typically proceeds through three stages (Ziegler et al., 2019; Rafailov et al., 2023): supervised fine-tuning (SFT) to establish basic instruction-following capabilities, reward modeling (RM) using the Bradley-Terry (BT) model to learn human preferences from pairwise comparisons, and reinforcement learning via algorithms like PPO (Schulman et al., 2017) to maximize rewards while maintaining a KL-divergence constraint to prevent distribution drift and reward hacking (Skalse et al., 2022; Miao et al., 2024). While effective, the high cost of human labor and the limited scalability of manual annotation have catalyzed the shift toward reinforcement learning from AI Feedback (RLAIF) (Lee et al., 2023) and Constitutional AI (Bai et al., 2022b). These methods leverage powerful off-the-shelf LLMs and a set of written principles to generate preference labels or revisions automatically.

To simplify the alignment pipeline, direct preference optimization (DPO) (Rafailov et al., 2023) reparameterizes the RLHF objective, allowing for stable and computationally lightweight training directly from preference data without an explicit reward model or an reinforcement learning phase. Despite its success, DPO can be vulnerable to overfitting and distribution shifts between the reference policy and the optimal policy. Identity preference optimization (IPO) (Azar et al., 2024) and $\Psi$-Preference optimization ($\Psi$PO) (Azar et al., 2024) framework address these theoretical limitations by employing bounded mapping functions to preserve the effectiveness of KL regularization, even when preferences are deterministic. Furthermore, to mitigate the distribution gap in offline datasets, online AI feedback (OAIF) (Guo et al., 2024a) and iterative preference learning (such as iterative DPO (Tu et al., 2025) and RSO (Liu et al., 2024)) employ on-policy sampling and real-time feedback during the training process, which has been shown to improve exploration and final performance.

## B.2. Modeling Preferences Beyond Binary Comparisons

**Unary Perference.** Recent research has explored alternative paradigms to reduce the data collection barrier and accommodate the complexity of human feedback. Kahneman-Tversky optimization (KTO) (Ethayarajh et al., 2024) draws from prospect theory to maximize the utility of generations directly from unpaired binary signals—categorizing outputs simply as desirable or undesirable. While KTO significantly lowers the bar for data collection and handles feedback intransitivity better than DPO, it focuses on isolated desirability rather than the comparative resolution between response candidates that is central to preference-based alignment.

**Ternary Perference.** To the best of our knowledge, TODO (Guo et al., 2025) represents the only prior effort to explicitly incorporate tie information into the alignment process. TODO introduces the tie-rank oriented Bradley-Terry (TOBT) model, an extension of the binary BT framework that accommodates a ternary ranking system: prefer, disprefer, and tie. By modeling ties, TODO prevents biased updates on indistinguishable responses, thereby enabling the model to learn from a broader spectrum of high-quality information that would otherwise be discarded or incorrectly labeled in a binary forced-choice setting.

Table 10. Alignment performance across different preference learning algorithms on UltraFeedback ($\delta_{\mathcal{M}} = 3$).

| Preference Method | Golden | Forced | Filtered | SymTie | D-KTO | C-DirTie |
|---|---|---|---|---|---|---|
| DPO | 6.48 | 7.94 | 8.29 | 8.39 | **9.92** | 10.28 |
| CPO | 5.37 | 5.84 | 6.24 | 6.33 | 7.45 | **7.96** |
| SimPO | 5.52 | 6.15 | 6.18 | 6.37 | 7.36 | **7.76** |

Table 11. Alignment performance using labelers trained with varying ratios of forced-choice tie noise.

| Tie Ratio | Forced | Filtered | TODO | SymTie | D-KTO | C-DirTie |
|---|---|---|---|---|---|---|
| 0% | 7.65 | 7.98 | 9.03 | 8.91 | 9.46 | 9.47 |
| 20% | **7.92** | **8.41** | **9.94** | **9.49** | **10.19** | **10.90** |
| 40% | 7.74 | 8.12 | 9.75 | 9.29 | 10.01 | 10.03 |

**Discussions** While TODO is the first to incorporate ties in preference learning, our work diverges from and extends this paradigm, serving as both a conceptual foundation and a methodological deepening of silent-aware alignment.

First, while TODO focuses on the optimization of tie supervision when they are already available, our primary contribution lies in *justifying why we need tie supervision in preference learning*. We demonstrate that a modest labeler (e.g., Llama-3B) can provide oversight that surpasses even human experts or forced frontier models by opting for silence on indistinguishable response pairs. This evidence elevates this silent-aware paradigm from a mere labeling alternative to a highly feasible and cost-effective trajectory for robust RLAIF.

Second, we deepen the silent-aware framework by advancing beyond the neutral ties modeled in TODO. Because our analysis confirms that weak supervisors can provide high-quality oversight when the forced-choice pressure is removed, we propose a more comprehensive labeling protocol. We demonstrate that besides neutral indistinguishability, annotators can extract quality-attributed signals at virtually zero additional computational cost. While these signals offer potent supervision, they remain unexplored in the current preference alignment literature, including TODO.

### B.3. Weak Supervision Paradigms: From Capability Elicitation to Teacher-Self Refinement

**Weak-to-Strong Generalization: Teacher vs. Student Elicitation.** Recent research has focused on the challenge of supervising superhuman models using weaker signals, a problem known as weak-to-strong (W2S) generalization (Burns et al., 2024; Shin et al., 2025; Lang et al., 2024; Yang et al., 2025b). Burns et al. (2024) proposed a paradigm where a strong pretrained model is finetuned on labels generated by a much weaker supervisor. The core hypothesis is that since strong models already possess task-relevant latent knowledge, the role of a weak supervisor is to elicit these capabilities rather than teach new ones. While experiments in NLP classification, chess, and reward modeling show that strong students consistently outperform their weak supervisors, naive finetuning often fails to recover the full potential of the strong model due to the risk of imitation saliency, where the student mimics the supervisor's systematic errors. In this paradigm, the central question is one of extrapolation: *can a "60-point" weak teacher successfully guide a "100-point" student to reach its own latent performance ceiling* without mimicking the teacher's flaws?

**Generative Alignment with Weak Teacher: Teacher vs. Teacher Replacement.** In contrast, Tao & Li (2025) shift the concern from the student vs. teacher gap to teacher vs. teacher efficiency in the generative alignment space. Their objective is not merely eliciting latent student capability, but rather identifying a sustainable and cost-effective replacement for expensive human or frontier-model feedback. By demonstrating that a task-specific 125M parameter supervisor can provide feedback that matches or exceeds human-annotated data, they prove that *the teaching capability of a "60-point" weak teacher can rival a "100-point" frontier teacher* in the context of scalability. In their view, the bottleneck is the cost of annotation, and the solution is the industrial-grade performance of task-specific small models, where supervisor scale has a minimized impact on alignment efficacy.

**The Silent-Aware Framework: Teacher vs. Self Refinement.** While Tao & Li (2025) establish that a weak teacher is a capable replacement, we argue that *its true potential is unlocked only when it is permitted to remain silent on indistinguishable response pairs*. We identify that a fundamental bottleneck in both W2S generalization and the weak teacher framework is the forced-choice constraint. Our work serves as a foundational deepening of the weak teacher framework by shifting the

*Table 12.* Alignment performance integrated with RE-PO dynamic latent variable estimation on UltraFeedback.

| Method | Confidence Interval | Gold Reward |
|---|---|---|
| w/ DPO (baseline) | N/A | 7.99 |
| w/ DPO + Filtered | [0.45, 0.55] | 8.08 |
| w/ DPO + Filtered | [0.40, 0.60] | 7.89 |
| w/ DPO + SymTie | [0.45, 0.55] | **8.13** |
| w/ DPO + SymTie | [0.40, 0.60] | 8.06 |
| w/ CPO (baseline) | N/A | 7.76 |
| w/ CPO + Filtered | [0.45, 0.55] | 7.78 |
| w/ CPO + Filtered | [0.40, 0.60] | 7.71 |
| w/ CPO + SymTie | [0.45, 0.55] | **8.01** |
| w/ CPO + SymTie | [0.40, 0.60] | 7.87 |
| w/ SimPO (baseline) | N/A | 7.98 |
| w/ SimPO + Filtered | [0.45, 0.55] | 7.97 |
| w/ SimPO + Filtered | [0.40, 0.60] | 7.86 |
| w/ SimPO + SymTie | [0.45, 0.55] | **8.05** |
| w/ SimPO + SymTie | [0.40, 0.60] | 7.93 |

objective toward high-fidelity supervision. While Tao & Li (2025) has also observed that both weak models and human experts exhibit significant uncertainty on subtle quality distinctions, their training protocol still requires the supervisor to guess a winner, which inevitably injects systematic noise into the gradient updates. Our framework introduces the honesty principle: the performance of alignment is governed more by the annotator's ability to recognize its own discriminative resolution limit than by its absolute model scale. In this light, we shift the comparative focus from inter-model gaps to the teacher vs. self paradigm: we demonstrate that when permitted to supervise in a honest manner, *a "60-point" weak teacher can significantly enhance its own teaching effectiveness*.

## C. Limitations and Future Work

While our study provides an in-depth investigation into the silent-aware alignment paradigm and reveals the fundamental role of the discriminative resolution limit, it is not without limitations that present exciting opportunities for subsequent research. We have established the efficacy of a static resolution limit for various labelers, the potential for dynamically adaptive thresholds that adjust to prompt complexity represents a sophisticated next step that we aim to address in future developments. Additionally, the rigorous theoretical underpinnings of these findings, particularly a formal information-theoretic proof of the regularization effects of silence, remain an area for further mathematical exploration.

As we look toward the future of scalable oversight, several strategic recommendations emerge from our study to further refine alignment methodologies. First, the success of our framework suggests that the field should move toward hybrid feedback systems. Integrating human nuanced insights with the consistent resolution limits of AI labelers could leverage the strengths of both, creating a supervision signal that is both qualitatively rich and statistically high-fidelity. Secondly, exploring the formalization of model honesty is essential. By extending our framework to include ethical constraints and multi-objective optimization, the field can transition toward more reliable and ethically responsible RLAIF strategies. By reclaiming the latent value in existing data and respecting the cognitive boundaries of supervisors, we can support the safe and efficient integration of increasingly capable AI systems into society.

