# OpenReview forum: "When Labelers Stay Silent: The Power of Ties in Cost-Effective Preference Learning"
_ICML.cc/2026/Conference — ICML 2026 regular_

### Official Review · Reviewer_b9Qa · 2026-03-09

**Soundness:** 3
**Presentation:** 3
**Significance:** 3
**Originality:** 2
**Overall Recommendation:** 4
**Confidence:** 3

**Summary:**

This paper studies an important issue in preference learning: many response pairs are inherently difficult to distinguish, and forced-choice supervision may introduce systematic noise. To address this, the authors propose a silent-aware framework that identifies tie-like pairs and designs several training objectives to better utilize them during alignment. Overall, the paper is well motivated and empirically promising.

**Compliance With Llm Reviewing Policy:**

Affirmed.

**Final Justification:**

The rebuttal addressed my main concerns and reinforced my prior assessment.

**Key Questions For Authors:**

My main concern is the first weakness and I refer the authors to that.

**Limitations:**

Yes.

**Strengths And Weaknesses:**

# Strength
1. A major strength of the paper is its motivation. It highlights a realistic yet often overlooked issue in preference learning: many response pairs are inherently hard to distinguish, and forced-choice supervision may introduce systematic noise. Framing this through the labeler’s discriminative resolution limit is intuitive and gives the paper a clear conceptual contribution.

2. Another strength is the completeness of the method and experiments. The paper goes beyond simply identifying ties, and proposes several silent-aware training objectives with fairly thorough empirical evaluation. This makes the work feel well-developed and practically meaningful.

# Weakness

1. I really appreciate the fairly comprehensive algorithmic design of the paper, including the diverse training objectives. My main concern, however, comes from the first phase, i.e., labeler training. As I understand it, the labeler is trained on the full preference data, which already contains choice-forced tie samples. In other words, the supervision used to train the labeler may itself be distorted by forced decisions on inherently ambiguous pairs. The subsequent pipeline then relies on this labeler together with a hard threshold to decide whether a sample is a tie. Although this design works well empirically, it does not seem fully reliable from a modeling perspective. In related work on alignment with noisy feedback, uncertainty is sometimes treated more explicitly as a latent variable and learned jointly, which feels more natural. For example, *Latent Collective Preference Optimization: A General Framework for Robust LLM Alignment* formulates label uncertainty as a latent-variable problem and uses EM-style estimation. I am not asking the authors to compare against that line of work directly; rather, I would hope to see an experiment that better validates the robustness of the proposed labeler itself. In particular, evaluating the labeler under different tie ratios or different levels of ambiguity would provide much stronger support for the overall pipeline.

2. I also appreciate the paper’s motivation from psychology; that intuition makes sense to me and is one of the appealing aspects of the work. That said, I think the paper would be even more impressive if the model design were connected more explicitly to prior choice models that already allow ties. I understand that for an application-oriented paper this may not be essential, so this is not my main concern. My primary concern is still the first point above, and I look forward to the authors’ response on that issue.

Cao, X., Xu, Z., Guang, M., Long, K., Bakker, M. A., Wang, Y., & Yu, C. (2025). *Latent Collective Preference Optimization: A General Framework for Robust LLM Alignment*. arXiv preprint arXiv:2509.24159.

---

> ### Author Rebuttal · Authors · 2026-03-31
>
> We would like to thank the reviewer for taking the time to review our work and the positive comments regarding its significance and experiments. Below are our responses to the comments in Weaknesses.
>
> > W1. The robustness of labelers.
>
> Thank you for your insightful suggestions. Your understanding of our labeler training is correct: it is trained on a small subset of forced-choice data. We do this to intentionally simulate a weak labeler, characterized by its small scale and its training on a noisy dataset. This setup further highlights our core finding: when ties are permitted, feedback from a weak labeler can match or exceed strong models. Following your suggestions, we validated its robustness under varying tie ratios and adapted an EM-style framework.
>
> First, we synthesized training sets with 0%, 20%, and 40% tie noise using 1/8 of the total dataset. From UltraFeedback, we select preference pairs with a GPT-4 score gap >= 3, treating them as clean preference data, and randomly replaced 20% and 40% of them with tie data (identical scores but assigned random forced-choice labels). After training proxy labelers on these sets, we annotated the full dataset using a uniform threshold $\delta_{\mathcal{M}}=3$ and trained the policy models.
>
> | Tie Ratio | Forced | Filtered | TODO | SymTie | D-KTO | C-DirTie |
> |-|-|-|-|-|-|-|
> |0% |7.65|7.98|9.03|8.91|9.46|9.47|
> |20% |7.92|8.41|9.94|9.49|10.19|10.90|
> |40% |7.74|8.12|9.75|9.29|10.01|10.03|
>
> Surprisingly, the policy models utilizing the noise labelers outperformed the clean baseline. We analyze why labelers trained with preference labels can express ties reliably. Clean labeler trained on easily distinguishable pairs, became overconfident and rarely output the small reward margins. In contrast, exposing the labeler to forced-choice noise on indistinguishable pairs provides inherently random and contradictory signals. Neural networks naturally learn generalizable patterns before overfitting to noise [1]. A model with limited capacity struggles to memorize these contradictory signals, so it remains unable to confidently distinguish ambiguous pairs after training. Therefore, applying a hard threshold is *not a mere empirical compromise, but a reliable and computationally free proxy. We appreciate this valuable suggestion and will include these results in the revised version.
>
> We fully agree that inferring sample uncertainty via EM-style estimation is theoretically rigorous, however, such approaches **still rely on** thresholds to determine which confidence interval represents a tie. Following your suggestion, we adapted RE-PO [2] to our setting. Instead of using a separately trained proxy labeler to explicitly annotate ties, we conducted experiments directly on the original dataset and employed RE-PO's posterior probability $w$ to identify ties on-the-fly. We set two confidence intervals for ties. Within each batch, samples falling into these ranges were processed using our proposed tie losses. The results show that our approaches consistently improve the performance over the RE-PO baselines.
>
> |RE-PO w/|Baseline|Filtered [0.45, 0.55] | Filtered [0.4, 0.6] | SymTie [0.45, 0.55] | SymTie [0.4, 0.6] |
> |-|-|-|-|-|-|
> |DPO|7.99|8.08|7.89|8.13|8.06|
> |CPO|7.76|7.78|7.71|8.01|7.87|
> |SimPO|7.98|7.97|7.86|8.05|7.93|
>
> > W2. Connect model design to prior choice models.
>
> Thanks for this constructive suggestion. Upon investigation, we found that SymTie shares a mathematical connection with the classical Rao-Kupper model [3].
>
> The Rao-Kupper model introduces a threshold $\delta > 0$, positing a tie occurs when the reward difference $\mu = r_1 - r_2$ in $[-\delta, \delta]$. This perfectly aligns with our discriminative resolution. Under the Rao-Kupper assumption, the probability of a tie is $P(y_1 \equiv y_2) = \frac{1}{1 + e^{-\mu - \delta}} - \frac{1}{1 + e^{-\mu + \delta}}$. By simplifying this probability and minimizing the negative log-likelihood, we obtain $\mathcal{L} = -\log(e^{2 \delta} - 1) + \log(1 + e^{\mu+\delta}) + \log(1 + e^{-\mu+\delta})$. Since $\delta$ is a constant hyperparameter, dropping the first term during optimization leaves: $\mathcal{L} = -\log \sigma(-\mu-\delta) - \log\sigma(\mu-\delta)$. This exactly matches our $\mathcal{L}_{\text{symTie}}$. Thank you and we will explicitly formalize this derivation in the revision to strengthen our theoretical foundation.
>
> [1] A Closer Look at Memorization in Deep Networks. ICML’17.
>
> [2] Latent Collective Preference Optimization: A General Framework for Robust LLM Alignment. arXiv preprint arXiv:2509.24159.
>
> [3] Ties in Paired-Comparison Experiments: A Generalization of the Bradley-Terry Model. Journal of the American Statistical Association, 1967.

---

> > ### Author Rebuttal · Reviewer_b9Qa · 2026-04-01
> >
> > Thanks for the responses, I maintain my positive score for this work.

---

> > > ### Author Response · Authors · 2026-04-02
> > >
> > > Thank you again for your time and support!

---

### Official Review · Reviewer_Gqgu · 2026-03-10

**Soundness:** 2
**Presentation:** 2
**Significance:** 3
**Originality:** 3
**Overall Recommendation:** 4
**Confidence:** 3

**Summary:**

This paper provides evidence that several datasets contain a significant proportion of pair-wise comparisons (judged as chosen/rejected) that are actually ties. By relying on *individual scores* of the elements being compared, provided by a golden labeller, the authors propose to partition the dataset into a group of ties (when the score difference is small), and actually preferred pairs (when the score difference is large). Then, authors propose extensions of the standard DPO loss that explicitly handle such ties, depending on their average score (good-good, and bad-bad). Finally, authors provide extensive empirical evidence that support that the losses they propose improve on DPO and the state-of-the-art.

**Compliance With Llm Reviewing Policy:**

Affirmed.

**Final Justification:**

I thank the authors for the helpful and constructive answers. Authors have addressed most of my points, notably settling terminology issues, and positioning relative to other fields (eg probabilistic modeling in social choice theory). One point, R3 - treatment of both-bad pairs, remains beyond my understanding. For these reasons, I stand by my weak accept score.

**Key Questions For Authors:**

The main questions follow the above items.
1. What does alignment performance means in your experiments?
2. Could you discuss the difference between your framework and that of classical DPO and social choice theory, and provide motivation for this change?
3. Why insist on a small margin for good-good pair, but a large margin for bad-bad pair? The reasoning at top of column 2 is unclear. Why would "a state where two poor responses have same score be a local minimum", since both scores could move while being equal?
4. What are definitions of notions of Loss and Risk, and their difference, in your work?

Some further minor aspects:
- Many terms are vague in the discussion:
  + l. 199: the unbiasedness of the learning objective. What would the bias be?
  + l. 172: counteracting the effects of negative learning. What does counteracting means here?
- You mention that "Bradley-Terry assumes definitive preference" between items. How is that so, since Bradley Terry is a probabilistic model?
- also at l. 145-147: "popular frameworks assume that definitive preferences exists for any two responses". This is a bit vague, could you specify your point and provide supporting citations?
- clarity: the notation $h_\theta$ in l. 205-208 and is the computation of $L_{symtie}$ is ambiguous.

**Limitations:**

yes

**Strengths And Weaknesses:**

### Strength
1. The exposition is mostly clear and easy to follow.
2. The existence of ties in existing datasets is convincingly supported by numerical experiments
3. The figures convincingly support that the proposed variants of DPO that leverage existence of ties improve performance over classical tie-unaware DPO -- under the caveat of Weakness 3.

### Weaknesses
1. Tables 1, 2, 3 & Fig. 2 report the "Alignment performance", but this term is not defined precisely. This casts a big shadow on the empirical evaluation of the work, which would otherwise be strong.
2. The work relies on the somewhat implicit, but quite important, assumption that the dataset of preferences writes as $(a, b, s_a, s_b)$ where $a$ and $b$ and the two elements (prompt/answer) being compared, and $s_\cdot$ being their respective individual scores. Standard DPO, along with a significant portion of the social choice theory (including Bradley-Terry and Packett-Luce models), assumes that the dataset writes as $(a, b, c)$, where $c$ is value that compares $a$ and $b$, typically either -1 or 1 for Bradley-Terry models (including DPO). This difference sets this work apart from the standard formulation of preference learning.
3.  Motivation of Directional Tie Loss is not convincing (p. 5 l. 228-231). I do not get the motivation of the Directional Tie Loss for bad-bad pairs , and specifically why scores of bad-bad pairs should be incentivized to be different, when scores of good-good pairs should be incentivized to be equal.
4. The notions of Loss and Risk (p. 5, including last sentence) are used interchangeably in the text, while classical statistical learning books give them specific meanings (see e.g. https://mitpress.mit.edu/9780262049443/learning-theory-from-first-principles/).

---

> ### Author Rebuttal · Authors · 2026-03-31
>
> We thank the reviewer for taking the time to review our paper and positive comments. Below are our responses.
>
> > W1/Q1. What does alignment performance means?
>
> Our evaluation metrics are detailed in the "Evaluation Metrics" paragraph of Sec 4.1. Specifically, in Tables 1, 2, 3 and Figure 2, it refers to the Gold Reward scored by Skywork/Skywork-Reward-V2-Llama-3.1-8B.
>
> > W2/Q2. Discuss the difference between your framework and classical DPO and social choice theory.
>
> We respectfully clarify that we **make no implicit assumptions** about the data format. The underlying mechanism for generating labels **aligns exactly** with standard DPO and social choice theory, as both rely on implicit or explicit scores to determine preferences. The apparent difference should stem from the fact that algorithm-focused methods typically treat pre-computed binary labels as a given starting point, whereas we shift the perspective upstream to formalize the data generation and labeling process. Explicitly formalizing this process is exactly what unlocks our key finding: a cost-effective model, when permitted to express ties, can provide alignment supervision that rivals or surpasses forced-choice frontier models.
>
> The primary distinction and core motivation of our paper is that, standard paradigms rigidly enforce binary labels $c \in \\{1,-1\\}$ even when score differences are negligible, whereas we introduce a "tie" label $c = 0$ when differences fall below a labeler's resolution limit. Our framework can be fully applicable to existing datasets, as demonstrated by our "Golden" baseline and supplementary experiments in our response to Reviewer b9Qa (W1), which also support the versatility of our work.
>
> > W3/Q3. Why insist on a large margin for bad-bad pair?
>
> Thank you for the comment. First, we must clarify that we do NOT insist on a large margin for bad-bad pairs. Although $-\mathcal{L}\_{\text{symTie}}$ creates a repulsive force at $h\_\theta = 0$, the loss function itself remains **strictly symmetric** with respect to $y_1$ and $y_2$. Flipping the sign neither designates a winner nor forces an asymmetrical margin, but merely signals that the current state of equal rewards is mathematically unstable, preventing the model from settling for low-quality outputs. Furthermore, this loss works synergistically with $\mathcal{L}\_{\text{marginDPO}}$, which globally provides a clear gradient toward high-reward regions. Under the dual pressure of this global guidance and the local repulsive force, the most efficient optimization path is NOT to make the probability of $y_1$ far greater than $y_2$, but to decrease the absolute probability mass of bad responses relative to the reference model. We will incorporate this clarification into the revised manuscript
>
> > W4/Q4. Definitions of Loss and Risk.
>
> We apologize for the confusion on p. 5. That is because the formulations of SymTie and D-KTO were primarily discussed at the point-wise loss level, while C-DirTie necessitated the risk level to introduce the corrected risk estimator, we used the umbrella term "objectives" for brevity to unify the methods defined in different levels. We will unify the expression style. In adddition, we have carefully checked the entire manuscript and we ensure these two terms are not actually used interchangeably. A loss function calculates the error for a single data point, whereas risk represents the expectation of that loss over a dataset. Our mathematical formulation strictly adheres to this convention.
>
> > Q6 & Q7. Clarification on "popular frameworks" and the BT model's assumption.
>
> We respectfully direct the reviewer to l. 7-16 in the Introduction, where we explicitly specify popular frameworks and provide the corresponding citations. We refer to dominant alignment paradigms RLHF and DPO. This assumption of definitive preference is enforced from two angles: practically, standard labeling protocols rigidly require selecting a definitive winner; theoretically, these frameworks rely on the standard BT model.
>
> Being a probabilistic model does NOT contradict the BT model assuming a definitive preference. It is strictly binary in its state space: $P(y_1 \succ y_2) + P(y_2 \succ y_1) = 1$. By forcing strict win and loss probabilities to sum to 1, it assumes a definitive winner always exists. Crucially, an output probability of 0.5 does NOT denote a tie; it merely denotes a 50% chance of either response strictly winning.
>
> > Q5 & Q8. Minor clarifications.
>
> Q5: We apologize but we could not locate the phrases "the unbiasedness of the learning objective" and "counteracting the effects of negative learning" in the manuscript. We would be glad to address these if a precise reference is provided. Q8: We will reformat all inline mathematical computations into dedicated equation environments to resolve any reading inconvenience. Thank you.

---

> > ### Author Rebuttal · Reviewer_Gqgu · 2026-04-02
> >
> > I thank the authors for the response. I have remaining comments / disagreement points with the authors:
> > - Q2. DPO and other social choice theory methods such as Bradley Terry do not require scores, but simply a binary comparison information. Many datasets come in this "chosen/rejected" format: HH-RLHF (Human preference data about helpfulness and harmlessness), PKU-Alignment/PKU-SafeRLHF, Nectar (2023) only has ranking information which is qualitatively different from scores. Your method consists in collecting scores for each element of the alternative, and mapping them into chosen/rejected/tie-both-good/tie-both-bad. This is qualitatively different from collecting comparison information directly. That being said, I get that your position is to propose an annotation pipeline.
> > - W3/Q3: thank you for the clarification. I am still somewhat confused by the fact that both-good pairs have an incentive to have equal scores, through Lsymtie, while both-bad pairs have an incentive to have different scores, through -Lsymtie.
> > - W4/Q4: I disagree. For instance, the objects of equations 8 and 9 are present as "losses", but are not pointwise relative to one datapoint. Rather, they concern the entire population.
> > - Q6&Q7: There is a misunderstanding on my question. Let me expand a bit. The BT model allows for two items $y_1$ and $y_2$ to have the some score, which I see as a non-definitive preference situation. Yet this a perfectly fine situation, where one can observe the random event "$y_1$ chosen" a number of times and learn that the scores of $y_1$ and $y_2$ are equal.  This is why I am bothered by the sentence "Bradley-Terry assumes definitive preference". Q7 is in the same spirit.
> > - Q5: I apologize for the confusing comment, it is an error on my part.

---

> > > ### Author Response · Authors · 2026-04-03
> > >
> > > We sincerely thank you for supporting our work and for the continued rigorous and constructive discussion. Your insights significantly help improve the mathematical precision and clarity of our manuscript.
> > >
> > > R2. Thank you for the insightful comment. We appreciate the precise distinction. It is true that many classical datasets are purely binary and lack explicit scores. Our framework is motivated by the modern RLAIF paradigm, where binary labels are often collapsed from the internal scores of an AI annotator.
> > >
> > > R3. We appreciate the detailed inquiry. Because the loss remains strictly symmetric with respect to $y_1$ and $y_2$, it inherently does not favor either response. Therefore, our objective for Both-Bad pairs is not to force a large margin between them. Rather, our goal is to prevent the gradient from vanishing at this low-quality state. Synergistically, the global $\mathcal{L}_{\text{marginDPO}}$ objective pulls the model toward high-quality regions, decreasing the absolute probability mass of both bad responses.
> > >
> > > R4. We sincerely apologize for this oversight! During the previous revision, we neglected to correct the notations in the formulas themselves. In the final manuscript, we will strictly correct this error by either removing the expectation operator $\mathbb{E}$ to accurately define the pointwise loss, or changing the notation from $\mathcal{L}$ to $\mathcal{R}$ to properly represent the risk. Thank you very much.
> > >
> > > R6&7: We agree with your statistical interpretation that the latent scores in the BT model can be equal ($r_1 = r_2$), yielding a 0.5 probability. When we wrote "assumes definitive preference", we were referring specifically to the observation space (label space) rather than the latent reward space. In standard BT formulations and forced-choice datasets, the allowed observational events are strictly binary. The annotator is never provided with a mathematical state to explicitly output a tie. Thank you for pointing this out. To eliminate this ambiguity, we will revise the relevant descriptions accordingly. For example, the statement at lines 145-147 will be modified as follows:
> > >
> > > "In LLM alignment, popular frameworks, typically grounded in the Bradley-Terry model, rely on a binary observation space that assumes a definitive preference exists between any two responses, thereby overlooking the de facto resolution limits of labelers."

---

### Official Review · Reviewer_jZ3U · 2026-03-13

**Soundness:** 2
**Presentation:** 2
**Significance:** 2
**Originality:** 2
**Overall Recommendation:** 4
**Confidence:** 4

**Summary:**

This paper highlights a key limitation of existing preference-learning approaches: we usually assume that for each given pair, there exists a preference. But authors claim that this is usually not the case, formulate the problem, and then propose a solution to mitigate the issues associated with the key point.

**Compliance With Llm Reviewing Policy:**

Affirmed.

**Final Justification:**

I feel ignoring the distributional shift issue is not the ideal thing to do, but I see the value of current contributions.

**Key Questions For Authors:**

Please refer to the above discussion.

**Limitations:**

Yes

**Strengths And Weaknesses:**

- The problem is interesting.
- The authors started with a practical limitation that exists in existing datasets and provided evidence to motivate the problem, which is nice.

- If both the responses are equally similar, and preference is hard, then why doesn't it imply that we don't need preference anymore because more is good (who is generating those responses), or maybe we can sample new pairs for which collecting preference is easier in the first place? This might need to be resampled and won't work on the dataset that already has such preferences, but can that be a valid solution? Just augment the dataset with more samples where we know responses are sufficiently different?

- Now, if the label has a region in terms of the reward model where it cannot label, then does the preference modelling assumption of equation (1) still hold?  Or in other words, does r* even exist, because if it does, then ideally, any two y's should be distinguishable because r* is continuous.

- I am not sure if the reward difference being more than delta is then justified to model the problem at hand. It seems that, in such cases, the underlying reward model needs to change. Any clarifications on that?

- Again, if the reward equation in (6) holds, it would amount to two different values for two different y's unless they are exactly the same, hence distinguishable?

- Can the authors report the KL divergence as well to the base model? It is important to understand if the reward gain is coming from better alignment or larger KL.

- The authors talked about a rigorous foundation, but the mathematical treatment is somewhat handwavy in the paper. Eventually, what it results in is changing the preference dataset?

- The conceptual contributions are incremental as compared to the TODO paper. The authors say that justification is the primary contribution, which is not convincing enough.

- Also, the key points in the framework are the threshold delta, which again seems to be chosen as a heuristic in practice. How is it selected?

---

> ### Author Rebuttal · Authors · 2026-03-31
>
> We would like to thank the reviewer for taking the time to review our work and acknowledging our motivation. Below are our responses.
>
> > W1. If both the response... different?
>
> We must clarify the premise and purpose regarding alignment paradigm seem to be conflated. Datasets are generated by external models and they can generate good responses does **not** imply that target model has mastered this capability. Alignment aims to distill this capability into the target model.
>
> Resampling might be valid under ideal conditions, but it is often **detached from the realities** of modern LLM engineering. High-quality data is notoriously scarce and expensive to acquire, and forcing the generation of sufficiently different equires a continuous generate-evaluate-discard loop, multiplying inference costs exponentially. Moreover, deliberately prompting models to produce different responses risks hallucinations or distribution shifts. Discarding tie data wastes powerful regularization signals (e.g., preventing over-optimization for stylistic deltas).
>
> > About reward. **W2**. Now, if... **W4**. Again, if...
>
> It is crucial to distinguish between objective latent variables and cognitive limits. While $r^\*$ is continuous and exact ties are mathematically improbable, **numerical differences cannot guarantee practical distinguishability** as neither human nor models possesses the infinite precision. This discrepancy motivates us to introduce the discriminative resolution limit. When differences fall below a discriminative resolution limit, the labeler naturally outputs a tie. This mechanism is also consistent with the classical Rao-Kupper model [1] (more details please refer to response to Reviewer b9Qa (W2)).
>
> > Role of $\delta$. **W3**. I am not sure... **W8**. Also, the key...
>
> $\delta$ is **not a hyperparameter** requires being "chosen" or "selected", but an intentional mechanism to simulate discriminative limits. By varying both $\delta$ and the proxy model scale, we are able to solidly validate our findings. As demonstrated in our experiments, SOTA models and humans both exhibit these limits. Ambiguity is an intrinsic, objective property of data, not merely a reward model deficiency. Therefore, changing the reward model *cannot* resolves the fundamental issue, and *contradicts* our main contribution: respecting a labeler’s resolution limit is more critical than enhancing its capability.
>
> > W5. KL divergence.
>
> We provide the KL divergence for the Llama-3-8B policy model corresponding to Table 2. To provide a reference point, the KL divergence between the base model and its Instruct counterpart is 301.82. These results demonstrate that our performance gain does not originate from a larger KL. In addition, Tables 4 and 8 can address reward hacking via OOD benchmark, proving our methods enhance abilities without degradation.
>
> |Labeler|$\delta$|Forced|Filtered|TODO|SymTie|D-KTO|C-DirTie|
> |-|-|-|-|-|-|-|-|
> |Llama-3B|1|6.83|8.10|8.28|8.11|12.31|12.56|
> ||3|6.83|9.62|13.13|10.61|17.83|26.98|
> ||5|6.83|10.53|15.77|11.19|19.44|25.36|
> |Llama-8B|1|7.09|8.86|9.49|9.42|13.00|13.40|
> ||3|7.09|9.52|19.30|16.70|22.83|26.52|
> ||5|7.09|10.19|26.87|19.87|27.57|35.98|
>
> > Novelty and contributions. **W6**. The authors... **W7**. The conceptual...
>
> We highlight that our work is fundamentally a **top-down framework shift** in preference learning, rather than an incremental extension of prior work or a trivial dataset modification. TODO treats ties merely as a third label, focusing on the *bottom-up* optimization step -- it assumes that tie supervision is already and always provided, which frequently fails in real-world scenarios. In contrast, our framework starts from practical observations of existing datasets and reconstructs the framework from the labeling paradigm *upwards*. We propose a complete pipeline for generating silent-aware data using low-cost models, where TODO is one specific algorithmic implementation that can be plugged.
>
> By constructing this top-down pipeline, we were able to **reveal a previously unrecognized phenomenon** in LLM alignment: respecting a labeler's resolution limit is significantly more critical than merely increasing its capability. This discovery establishes a highly cost-effective trajectory for robust RLAIF. We humbly believe that the significance and novelty of research should not be limited to algorithm design alone. Our findings and proposed strategy are expected to draw attention to silent-aware framework, which may *push forward the field of LLM alignment as a whole*.
>
> Regarding W6, our "rigorous foundation" refers to the conceptual basis for breaking forced-choice constraints, not heavy mathematical proof. Sorry for the confusion and we will clarity it.
>
> [1] Ties in Paired-Comparison Experiments: A Generalization of the Bradley-Terry Model. Journal of the American Statistical Association, 1967.

---

> > ### Author Rebuttal · Reviewer_jZ3U · 2026-04-02
> >
> > Dear Authors,
> >
> > Thank you for your response.
> >
> > The samples in equation 3 in the paper were generated by the model pi_\theta, but in equation 4, how can they come from some other model that is more capable? There seems to be a mismatch in understanding here. Because if the samples are coming from the other model (as the authors wrote "Datasets are generated by external models"), does a price of distribution shift need to be paid?
> >
> > Regards,
> >
> > Reviewer

---

> > > ### Author Response · Authors · 2026-04-03
> > >
> > > We sincerely appreciate your follow-up. To address the confusions regarding the standard alignment pipeline, dataset generation, and distribution shift, we clarify the foundational concepts below:
> > >
> > > 1. The transition from Eq 3 to Eq 4 is the **mathematical foundation of DPO**. Eq 3 is the theoretical RLHF objective, which assumes on-policy sampling from the policy model $\pi_{\theta}$. DPO reparameterizes this RL problem into an off-policy classification task (Eq 4). This allows the policy to be optimized using static, offline datasets.
> > >
> > > 2. **Whether** training a reward model $r$ in RLHF **or** optimizing a policy $\pi$ in DPO and its variants, the response candidates $(x, y_c, y_r)$ are **already** generated offline by some external models, similar to how images in classification datasets (e.g., ImageNet) are scraped before training. This is exactly why we stated "Datasets are generated by external models".
> > >
> > > 3. Just as ImageNet requires manual categorization, annotators in alignment tasks typically assign forced binary preference labels to these pre-collected pairs. Furthermore, just as a vision researcher cannot dynamically photograph new objects during training simply because an ImageNet label is noisy, dynamically resampling responses during offline alignment is unviable. Existing data is exceptionally expensive, and the consensus is to maximize the utility of them.
> > >
> > > 4. Dominant alignment paradigms like RLHF and DPO train strictly utilize binary datasets. We highlight that the systemic noise in current datasets partially stems from this forced binary constraint. Forcing annotators to declare a winner between indistinguishable pairs is loosely akin to forcing a continuous spectrum of subtle differences into a strict binary taxonomy. Our silent-aware framework intervenes exactly at this labeling stage.
> > >
> > > 5. We fully agree that using off-policy data incurs a "price of distribution shift". However, **this is a universal characteristic of all offline alignment methods**, not a flaw of our framework. In fact, offline paradigms like DPO are widely adopted exactly because they offer superior training stability and avoid the complexities of online RL. In short, this distribution shift is strictly orthogonal to our contribution. Our methods and all baselines are evaluated under the exact same standard setting.

---

### Official Review · Reviewer_4Lp1 · 2026-03-14

**Soundness:** 2
**Presentation:** 3
**Significance:** 2
**Originality:** 2
**Overall Recommendation:** 4
**Confidence:** 3

**Summary:**

This paper proposes to explicitly model the ties in preference data pairs, and proposes the preference optimization loss which takes into account the tie signals. Experimental results are provided to showcase the advantage over the baseline approaches like Forced or Filtered methods.

**Compliance With Llm Reviewing Policy:**

Affirmed.

**Final Justification:**

The rebuttal addressed some of my concerns.

**Key Questions For Authors:**

Q1: How is the paper connected to the existing works like "Less is More for Alignment"? Is the proposed methodology in the paper complementary to these methods or the proposed method in the paper requires certain non-perfectness of the data, like certain ties?

**Limitations:**

See weakness.

**Strengths And Weaknesses:**

**Strength:**

1. The study of processing annotation data is important and this paper provides some new perspectives in taking into account the significance of the perference pairs.

**Weakness:**

1. The comparison is only conducted on KTO; there lacks the study of applicability of the strategy to other existing preference alignment approaches like DPO, IPO, SimPO etc. This makes the performance gain of the methodology questionable. Will the gains remain valid for other preference optimization methods?

2. There are several existing DPO variants papers which has taken into account the preference significance, which makes the novelty of the paper rather limited.

---

> ### Author Rebuttal · Authors · 2026-03-31
>
> We would like to thank the reviewer for taking the time to review our work and acknowledging our new perspectives on data annotation. Below are our responses.
>
> > **W1**. The comparison is only conducted on KTO ... Will the gains remain valid for other preference optimization methods? **W2**. There are several existing DPO variants papers which has taken into account the preference significance, which makes the novelty of the paper rather limited.
>
> Thank you for the comments. We respectfully note that there may be some misunderstandings regarding our primary focus and the significance of experiments, which tightly connect W1 and W2. We would like to address these points holistically.
>
> Our motivation is to highlight that many response pairs are inherently hard to distinguish, and forced-choice supervision introduces systematic noise. We highlight the key *conceptual innovation and core finding* of our work is that respecting a labeler’s resolution limit is significantly more critical than merely increasing its capability.
>
> **R1.** We must clarify a factual error in W1: the comparison is **not** conducted on KTO. KTO is only used in D-KTO, where we innovatively apply KTO to process tie data with quality signals.
>
> To maintain a fair evaluation, the optimization algorithm for preference data across all methods was uniformly set to DPO. Because the *significance of our experiments* lies in revealing the power of the silent-aware framework with the proposed objectives for tie data, rather than the specific implementation details of how the preference data is processed, algorithms designed for preference data, like CPO and SimPO, are **orthogonal to our focus**.
>
> Following your suggestion, we added CPO and SimPO as alternative approaches for the preference data. The performance on UltraFeedback with feedback from the golden labeler GPT-4 and a weak labeler Llama-8B ($\delta_{\mathcal{M}}=3$) is presented below. All main results perfectly align with the core findings in our paper, and all these methods *show the same trend* as DPO does and *consistently benefit* from our proposed objectives.
>
> |Pref-method\\Tie-method|Golden|Forced|Filtered|SymTie|D-KTO|C-DirTie|
> |-|-|-|-|-|-|-|
> |DPO|6.48|7.94|8.29|8.39|9.92|10.28|
> |CPO|5.37|5.84|6.24|6.33|7.45|7.96|
> |SimPO|5.52|6.15|6.18|6.37|7.36|7.76|
>
> **R2.** Both our approach and preference significance (PS) methods are motivated by preference dataset noise, but their **perspectives and solutions are completely different**. PS methods process noisy data *after* collection. In contrast, our framework focuses on obtaining cleaner data *during* annotation. Notably, we found that relaxing the forced-choice constraint is more efficient and effective than employing a stronger model to force choices.
>
> To further validate our framework within the context of PS methods, we added experiments on a SOTA PS framework, RE-PO [1]. By applying our tie losses to samples falling into RE-PO's mid-range confidence interval which indicates ambiguity, our approaches consistently improved performance over the RE-PO baselines. This confirms that PS methods **cannot replace our framework's ability** to model indistinguishable pairs. For detailed experimental results, please kindly refer to our response to W1 of Reviewer b9Qa.
>
> > Q1. How is the paper connected to the existing works like "Less is More for Alignment" (LIMA)? Is the proposed methodology in the paper complementary to these methods or the proposed method in the paper requires certain non-perfectness of the data, like certain ties?
>
> LIMA's "quality over quantity" philosophy aligns with our results that Filtered outperforms Forced. However, LIMA, *much like PS methods*, focuses on downstream data selection, while our work answers a practical question at the upstream data collection phase: how can we achieve high-fidelity supervision cost-effectively. Consequently, these two paradigms synergize: our protocol provides a high-fidelity foundation for downstream methods to further curate. As shown in R2, downstream methods are **further elevated** when integrated with our upstream protocol.
>
> Crucially, our method does NOT "require" ties; rather, ties are an objective reflection of labelers' discriminative boundaries. Forcing a binary choice on indistinguishable pairs injects noise, and our framework allows us to **actively prevent**. Furthermore, as demonstrated in R2, our framework remains effective even without explicitly labeled ties.
>
> Finally, we respectfully believe that ties do NOT necessarily equate to data imperfection. For example, the "Both-Good" signal instructs the policy model to treat equally excellent candidates as equivalent. This prevents the model from wasting capacity by over-optimizing for negligible stylistic deltas. In this light, acknowledging ties is not a concession to bad data, but a regularization strategy.
>
> [1] RE-PO: Robust Enhanced Policy Optimization as a General Framework for LLM Alignment. ICLR'26.

---

> > ### Author Rebuttal · Reviewer_4Lp1 · 2026-04-02
> >
> > Thanks authors for ther response. Despite this, I still have questions on the difference between the paper's contribution and existing data curation method like LIMA. Overall, the literature has several papers working on preference data curating, and the current responses of the authors do not convince me of the new contributions. What are the differences in contributions?

---

> > > ### Author Response · Authors · 2026-04-03
> > >
> > > We appreciate the follow-up. To directly address the comparison: our framework and data curation methods like LIMA operate on fundamentally different levels of the alignment pipeline. LIMA is a downstream data curation strategy operating within the conventional forced-choice paradigm, whereas our work is a top-down framework shift that redefines the paradigm itself.
> > >
> > > The fundamental distinctions are as follows:
> > >
> > > 1. Methods like LIMA *passively* filter data downstream. However, they ignore a root cause: forcing binary choices systematically introduces noise. Our silent-aware framework breaks this convention. By removing the forced-choice constraint, we shift from reactive data filtering to **proactive, zero-cost noise prevention at the annotation source**.
> > >
> > > 2. By reconstructing the labeling protocol, our work is the *first to reveal a quite important phenomenon*: respecting a labeler's resolution limit is more critical than increasing its capability. The importance of our work is underscored by its potential to **address trade-offs between annotation costs and alignment quality**, making high-quality RLAIF more scalable and accessible --- a systemic insight that data filtering methods *inherently cannot* provide.
> > >
> > > 3. **Our framework empowers downstream filtering methods**. As shown in Tables 2-4, 8, and our first rebuttal, the silent-aware protocol consistently improves SOTA methods, including standard DPO and its variants (e.g., SimPO) and data filtering/reweighting methods (e.g., RE-PO). Furthermore, building on this foundation, our proposed objectives (SymTie, C-DirTie) push performance boundaries *even higher*.
> > >
> > > In fact, these differences perfectly encapsulate our core contributions. We will refine our discussion in the revised manuscript. Thank you.

---

### Decision · Program_Chairs · 2026-04-30

**Decision:**

Accept (regular)

**Comment:**

All reviewers gave this a 'weak accept' and had similar views: This paper addresses an important and overlooked question: how to properly use ties in preference data, and makes a solid and convincing contribution. The particualrs of the  loss formulation are novel and the empirical findings convinced the reviewers in general (though there were some complaints of how comprehensive comparisons were). The main drawback was on how strong the novelty was (given that there are other extensions to DPO that touch on similar topics, CPO etc). This is what seems to pull it back from a strong accept, but nevertheless a valuable contribution.